# Parameter Competition Balancing for Model Merging

**Guodong Du**[1*]   **Junlin Lee**[1*]   **Jing Li**[1†]   **Runhua Jiang**[2]   **Yifei Guo**[2]   **Shuyang Yu**[2]
**Hanting Liu**[3]   **Sim Kuan Goh**[2]   **Ho-Kin Tang**[1†]   **Daojing He**[1]   **Min Zhang**[1]

[1]Harbin Institute of Technology, Shenzhen, China
[2]Xiamen University Malaysia
[3]Johns Hopkins University

duguodong7@gmail.com   jingli.phd@hotmail.com
denghaojian@hit.edu.cn

## Abstract

While fine-tuning pretrained models has become common practice, these models often underperform outside their specific domains. Recently developed model merging techniques enable the direct integration of multiple models, each fine-tuned for distinct tasks, into a single model. This strategy promotes multitasking capabilities without requiring retraining on the original datasets. However, existing methods fall short in addressing potential conflicts and complex correlations between tasks, especially in parameter-level adjustments, posing a challenge in effectively balancing parameter competition across various tasks. This paper introduces an innovative technique named PCB-MERGING (Parameter Competition Balancing), a *lightweight* and *training-free* technique that adjusts the coefficients of each parameter for effective model merging. PCB-MERGING employs intra-balancing to gauge parameter significance within individual tasks and inter-balancing to assess parameter similarities across different tasks. Parameters with low importance scores are dropped, and the remaining ones are rescaled to form the final merged model. We assessed our approach in diverse merging scenarios, including cross-task, cross-domain, and cross-training configurations, as well as out-of-domain generalization. The experimental results reveal that our approach achieves substantial performance enhancements across multiple modalities, domains, model sizes, number of tasks, fine-tuning forms, and large language models, outperforming existing model merging methods. The code is publicly available at: https://github.com/duguodong7/pcb-merging.

## 1 Introduction

Pre-trained models (PTMs) are fundamental in deep learning, underpinning many current techniques due to their ability to learn generalized features from large datasets [99, 5]. Fine-tuning PTMs for specific tasks is a common practice to boost performance [71, 31]. This approach is prevalent, resulting in thousands of fine-tuned checkpoints [85], based on widely used PTMs [59, 80, 58]. However, fine-tuning the same model for different tasks can result in performance variations, posing a significant challenge [57]. Multi-task learning [66, 59] has been proposed as a solution, but it incurs substantial training costs and requires simultaneous access to data and labels for all tasks [16]. Recently, some researchers have developed methods to merge multiple independently-trained models into a single model without the need for original training data [20, 86, 27]. This merging technique not only adheres to data privacy regulations [83] but also enhances efficiency by eliminating the need for retraining.

---

[*] Equal contribution.
[†] Corresponding authors.

38th Conference on Neural Information Processing Systems (NeurIPS 2024).

Table 1: Comparison of different model merging methods. A merging method is deemed *self-aware* if it manages parameter competition within individual task models, and *cross-aware* if it balances competition within a population of task models. For more details, please refer to App. A.

| Method | Drop | Scale | Self-aware | Cross-aware | Granularity Level |
|---|---|---|---|---|---|
| Fisher Merging [NeurIPS22] [46] | - | Fisher Matric | ✓ | ✗ | Parameter |
| RegMean[ICLR23] [30] | - | Inner Product Matric | ✗ | ✓ | Parameter |
| Task Arithmetic[ICLR23] [28] | - | Uniformed | ✗ | ✗ | Task |
| TIES-Merging[NeurIPS23] [89] | Magnitude | Uniformed | ✓ | ✓ | Parameter |
| DARE[ICML24] [94] | Bernoulli ($p$) | $1/(1-p)$ | ✓ | ✗ | Parameter |
| LoraHub[COLM24] [25] | - | Evolver Searched | ✗ | ✓ | Task |
| AdaMerging[ICLR24] [90] | - | Unsupervised Optimized | ✗ | ✓ | Layer |
| **PCB-MERGING (ours)** | Competition | Balancing Matric | ✓ | ✓ | Parameter |

Previous research [20, 86, 27] has shown that averaging the weights of multiple task-specific models, fine-tuned from the same pre-trained initialization, can enhance performance across various tasks. Many studies [46, 30] have explored the creation of additional matrices, matching the model dimensions, to adjust parameter coefficients for different tasks. Other studies [28, 89, 90, 96, 94] focus on task vectors [28], defined as the differences between the parameter values of the fine-tuned model and the original pre-trained model. While these task vector-based methods have shown promising results, they typically apply a uniform coefficient for each task and parameter, which may limit their effectiveness. Our research seeks to fully harness task vector-based methods by fine-tuning parameter-level coefficients through a balancing mechanism that resolves parameter competition.

Parameter competition is crucial in model fusion, occurring both within parameters of the same task and among models for different tasks. Firstly, within a single model, task-specific fine-tuned parameters often compete, where some are critical while many prove redundant. Previous research [89, 94] has demonstrated that dropping numerous parameters based on task vector magnitude can maintain performance close to the original. Additionally, appropriately rescaling important parameters and suppressing redundant ones can further enhance the performance of the fine-tuned model (see Fig. 1). Secondly, between different models, parameters also engage in competition (see Fig. 2). Rescaling a task vector for one task can boost performance for that specific task but may negatively affect cross-task capabilities. Therefore, balancing the coefficients assigned to task vectors requires careful consideration of their impact on overall performance.

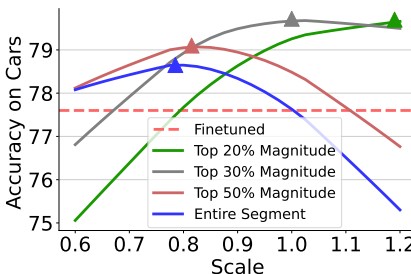

Figure 1: Parameter competition within individual task models. Intra-balancing enhances performance beyond finetuning.

We argue that merging methods capable of managing intra-parameter competition within tasks demonstrate *self-awareness*, while those that balance inter-parameter competition between tasks exhibit *cross-awareness*. We systematically compare and analyze existing model merging methods in terms of these criteria, as presented in Tab. 1. To establish a balancing matrix that is both self-aware and cross-aware for parameter scaling, we introduce PCB-MERGING (**P**arameter **C**ompetition **B**alancing for Model Merging), a *training-free* and *dataless* method for merging models. Specifically, we use intra-balancing to weight the importance of parameters within tasks and inter-balancing to assess parameter similarities across tasks. Low-scoring parameters are then dropped, and the remaining ones are rescaled. Finally, we merge the modulated task vectors into the pretrained model to create the final merged model.

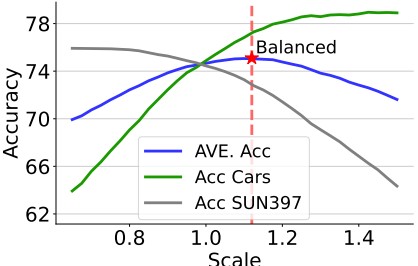

Figure 2: Parameter competition within task model populations. Inter-balancing improves cross-task generalization.

To empirically demonstrate the effectiveness of PCB-MERGING, we conducted extensive experiments comparing it with existing model merging approaches. We showcased the superiority of our

approach from four perspectives: (1) Cross-task merging: We evaluated our approach across a range of NLP and Vision tasks using various models, such as T5 [59], ViT [12], and Llama2 [80]. We also assessed its ability to fuse multiple PEFT [42, 24] adapters. All experiments demonstrated significant improvements over previous state-of-the-art methods, notably achieving a 4.3% performance increase with the T5-base model. (2) Cross-domain merging: Our approach merged multiple domain-specific models for tasks like emotion classification [53, 30], demonstrating its effective handling of diverse domain data. (3) Cross-training configurations: Merging multiple models from different training environments on single tasks, highlighting its flexibility and robustness. (4) Out-of-Domain Generalization: We assessed multi-task and multi-domain fusion performance on domain shift datasets, testing generalizability across various frameworks.

This paper makes three significant **contributions**: (1) We re-examine existing model merging methods, highlighting the critical role of parameter competition awareness; (2) We introduce a novel approach called PCB-MERGING, which effectively adjusts parameter coefficients through balancing parameter competition; (3) Our proposed method stabilizes and enhances model merging performance across various application scenarios without additional training.

## 2 Related Work

### 2.1 Overview of model fusion

Deep model fusion is gaining attention due to data privacy and resource conservation concerns, with potential applications across various domains [39, 14]. It's typically divided into three main categories. Ensemble learning [64], combines model outputs to improve prediction accuracy and robustness but requires parallel deployment of multiple models. An alternative method involves mode connectivity [18] and alignment [1], aiming to bring solutions closer together for better initial conditions in averaging. This is achieved by either linking optimization paths [15, 98] or addressing permutation invariances [72, 79, 40, 30] . Recent researches [88, 75] focus on training-free approaches to enhance model fusion usability. The third approach, weight averaging [20, 86], requires models with identical structures. While advancements like [81] support merging diverse large language models (LLMs), they require knowledge distillation [23] and complex training. This paper follows the third type of track due to its simplicity, efficiency, and broad applicability.

### 2.2 Merging fine-tuned models with same initialization

Previous studies found that when multiple models are fine-tuned from the same pre-trained initialization, averaging their weights can lead to improved performance on single tasks [20, 86, 13, 29, 92] different tasks [27] and out-of-distribution generalization [3, 60]. Fisher Merging [46] goes beyond simple averaging to identify the importance of individual parameters using Fisher information matrix [17] and uses it to weigh the parameters in each model when merging. RegMean [30] proposed a closed-form solution for the merged model's parameters by solving a local linear regression problem for each individual linear layer in the model. However, both the Fisher Merging and RegMean methods are time-consuming and computationally intensive.

Task Arithmetic [28] introduces the concept of *task vectors*, demonstrating their effectiveness and lightweight nature in facilitating cross tasks generalization. Expanding on this groundwork, PEM Composition [96] extends the task arithmetic framework to merge LoRA [24] models, while Ties-Merging [89] addresses task conflicts by resetting redundant parameters and resolving sign conflicts. However, these methods share a merging coefficient across all task vectors, limiting flexibility. In contrast, Lorahub [25] and AdaMerging [90] utilize different coefficients for enhanced adaptability, but Lorahub's performance is restricted as it only searches coefficients at the task level. AdaMerging also demands complex training and unlabeled test datasets and is applicable solely to classification problems. DARE [94] proposes drop and rescale as a preprocessing step when merging fine-tuned LLMs. Our approach primarily employs strategies of dropping to minimize interference and rescaling at the parameter level, while considering both self-awareness and cross-model awareness.

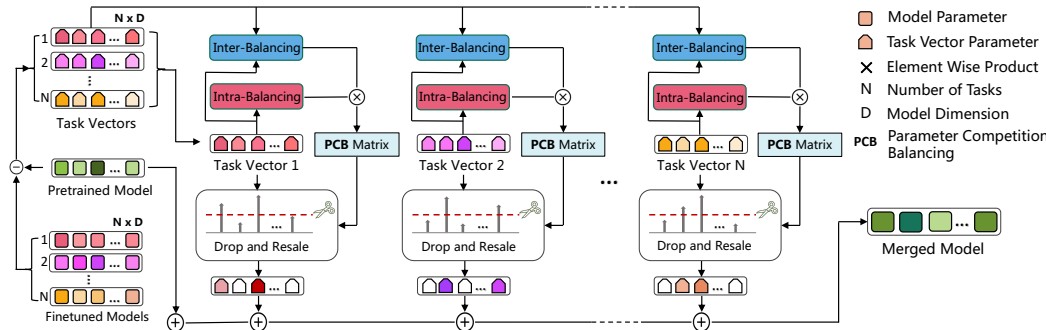

Figure 3: An illustration of the steps in PCB-MERGING. Different colored blocks represent parameters with varying values. We start with multiple fine-tuned models and a pretrained model, establishing a **PCB** matrix through intra-balancing and inter-balancing. Low-scoring parameters are dropped, and the remaining ones are rescaled. Finally, we merge the modulated task vectors into the pretrained model to create the final merged model.

# 3 Method

In Sec. 3.1, we established the notation and outlined the problem of model merging. Sec. 3.2 delves into the detailed exposition of the proposed PCB-MERGING method, which aims to balance parameter competition. Furthermore, in Sec. 3.3, we employ evolutionary algorithms to further enhance the performance of our approach.

## 3.1 Preliminaries

Initially, we are faced with a set of tasks $\{T_1, \ldots, T_n\}$ and various pre-trained models, such as ViT [12], T5 [59], or llama2 [80]. We have the option to fine-tune the entire model or employ a parameter-efficient fine-tuning (PEFT) method [42, 24]. During fine-tuning, we represent the trainable parameters as $\theta$, initialized as $\theta_{\text{pre}}$, and the fine-tuned parameters as $\theta_{\text{ft}}$. The model merging problem involves how to combine the weight sets $\{\theta_1, \ldots, \theta_n\}$ to form a new weight $\theta_m$, without the need to retrain using the initial training data for each task, and ensuring that $\theta_m$ can simultaneously perform tasks $\{1, \ldots, N\}$.

Recent research [28] introduced the concept of *task vectors* and completed various task arithmetic operations and model merging based on task vectors. Specifically, for task $T_i$, the task vector $\tau_i \in \mathbb{R}^d$ is defined as the vector obtained by subtracting the fine-tuned weights $\theta_i$ from the pre-trained weights $\theta_{\text{pre}}$, i.e., $\tau_i = \theta_i - \theta_{\text{pre}}$. This allows us to focus on the changes that occur during each task-specific model's fine-tuning phase. The task vector-based multi-task model merging method can be expressed as $\theta_m = \theta_{\text{pre}} + \lambda * \sum_{i=1}^n \tau_i$, where the coefficient $\lambda$ represents the importance of merged task vector $\tau_m$. This concept is simple yet effective, significantly outperforming simple weight averaging schemes, i.e., $\theta_m = (1/N) \sum_{i=1}^n \theta_i$.

## 3.2 Parameter Competition Balancing

Our approach aims to modulate the scaling factors for each task and parameter, achieving intra-balancing and inter-balancing within and between tasks. Specifically, we use the parameter competition balancing (PCB) matrix $\beta_i \in \mathbb{R}^d$ to adjust the scale of parameters in each task model $\theta_i \in \mathbb{R}^d$, resulting in the final fused model, as shown in Fig. 3. The specific calculation process is as follows:

1. **Intra-Balancing:** Initially, we implement self-awareness by applying a nonlinear activation function (i.e., softmax) to the magnitudes of task vectors, emphasizing important parameters while suppressing redundant ones to some extent. As the number of fusion tasks increases, competition among parameters intensifies. Therefore, the number of tasks $N$ is used to control the extent of suppression applied to redundant parameters. "Norm" refers to normalization.

$$\beta_{intra,i} = \text{Softmax}(N * \text{Norm}(\tau_i \odot \tau_i)) \tag{1}$$

2. **Inter-Balancing:** Next, we realize cross-awareness to enable the parameters within a population of tasks to interact with others, addressing potential conflicts and complex correlations between tasks. To achieve this, we compute the similarity between parameters at the same positions across different task vectors, allowing each parameter to update its score based on information from other tasks. The calculation process is as follows:

$$\beta_{inter,i} = \sum_{j=1}^{n} \text{Softmax}(\text{Norm}(\tau_i \odot \tau_j)) \tag{2}$$

3. **Drop and Rescale:** Subsequently, we obtain $\beta_i = \beta_{intra,i} \odot \beta_{inter,i}$. Next, we construct a mask $m_i \in \mathbb{R}^d$ based on $\beta_i$ to focus on the more important parameters. Specifically, this mask $m_i$ is used to select high-scoring elements from the $D$ elements of $\beta_i$. We define the mask ratio as $r$, where $0 < r \leq 1$. The mask $m_i$ can be derived from:

$$m_{i,d} = \begin{cases} 1, & \text{if } \beta_{i,d} \geq \text{sorted}(\beta_i)[(1-r) \times D] \\ 0, & \text{otherwise} \end{cases} \tag{3}$$

The importance score is defined as $\hat{\beta} = m_i \odot \beta_i$. Finally, we use the score of the masked balancing matrix to weight the importance of each parameter in each task vector. The final merged task vector $\tau_m$ is as follows:

$$\tau_m = \sum_{i=1}^{n} (\hat{\beta}_i \odot \tau_i) / \sum_{i=1}^{n} \hat{\beta}_i \tag{4}$$

From the final merged task vector $\tau_m$, we can further adjust its magnitude proportionally and integrate it with the initial parameter values to yield the amalgamated model parameters $\theta_m$, represented by $\theta_m = \theta_{\text{pre}} + \lambda * \tau_m$, with $\lambda$ serving as a scaling hyperparameter. More details about the method workflow are presented in App. A and Algorithm 1.

## 3.3 Searching Coefficients

Research from articles [28, 90] shows that model merging methods based on task vectors are highly sensitive to the merging coefficient $\lambda$. Even with an appropriately chosen uniform $\lambda$, achieving further improvements in fusion performance necessitates grid searching the merging coefficients for each task vector, which becomes increasingly complex and time-consuming, especially when managing a large number of tasks.

Inspired by prior research [77, 25], we employ intelligent optimization algorithms to search for mixing coefficients, aiming for greater improvements compared to using a uniform coefficient. The optimization process seeks the best set $\{\lambda_1, \ldots, \lambda_n\}$ to enhance validation accuracy, with the ultimate goal of maximizing validation accuracy with the merged model.

$$\theta_m = \theta_{\text{pre}} + \sum_{i=1}^{n} (\hat{\beta}_i \odot \lambda_i \tau_i) / \sum_{i=1}^{n} \hat{\beta}_i \tag{5}$$

In most of our experimental setups, we primarily utilize Covariance Matrix Adaptive Evolution Strategies (CMA-ES) [21]. As a probabilistic population-based optimization algorithm, CMA-ES dynamically adjusts the search distribution defined by the covariance matrix. It systematically updates the mean and covariance of this distribution at each iteration to learn and exploit the underlying structure of the search space for optimization efficiency.

## 4 Experimental setup

**Evaluation Settings.** We anticipate that merging models will offer two significant advantages for developers. Firstly, by integrating insights from individual models $\theta_{1..n}$ trained in different environments (such as tasks, domains, or various training configurations within a single task), we expect the resulting merged model $\theta_m$ to demonstrate competitive test performance across tasks, domains, or within a single task. Secondly, this merged model is poised to exhibit enhanced cross-domain (OOD) generalization capability. For further details about compute resources and fine-tuning procedures, please refer to App. F.1 and F.2.

**Baseline Methods.** Our baselines are primarily divided into two categories: non-model merging, which involves fine-tuned individual models and multitask learning, and various advanced model merging methods such as simple averaging [86], Fisher merging [46], RegMean [30], Task Arithmetic [28], Ties-Merging [89], and AdaMerging [90]. Detailed information on these baselines can be found in App. E. Notably, Task Arithmetic, Ties-Merging, AdaMerging, and our proposed PCB-MERGING method are all based on task vectors. In addition, when merging LLMs across different tasks, we present the results with DARE [94] as preprocessing. Since AdaMerging demands unlabeled test datasets and is applicable solely to classification problems, we compare with it only when merging finetuned ViT models for image classification, as shown in App. C.2.

**Validation Set.** Most model merging methods necessitate access to a validation set, utilized for computing the Fisher matrix or tuning hyperparameters. While ReMean can derive inner product matrices for each task using unlabeled training data, additional validation is required to ascertain the optimal value of the non-diagonal multiplier $\alpha$. Both Fisher merging and ReMean are time-consuming and require significant computational resources. In contrast, task vector-based methods are more lightweight and training-free to implement and can be utilized even without a validation set. Therefore, we conducted additional experiments to compare task vector-based methods without a validation set.

**Hyperparameters.** When no additional validation is performed, we use a default value of $\lambda = 1$ for all task-vector based methods. For TIES-Merging and PCB-MERGING, which require a masking ratio, we set mask ratio $r = 0.2$ as the default value for all experiments, except in LLM experiments where $r = 0.1$.

When validation is allowed, we set the non-diagonal multiplier $\alpha$ in RegMean to 0.9, except for the T5-base model where it is set to 0.1. For Task Arithmetic, we conduct a search over $\lambda$ ranging from 0.2 to 1.5 with a step size of 0.1. For TIES-Merging and PCB-MERGING, we search over ratios in $\{0.05, 0.1, 0.2\}$, and $\lambda$ ranging from 0.8 to 2.5 with a step size of 0.1. In cases where evolutionary strategies are employed for coefficient search for each task, we conduct continuous variable searches within the range of 0.8 to 2.5. For more hyperparameter details, please refer to App. F.3 and Tab. 17.

## 5 Results

In this section, we evaluated the performance of the PCB-MERGING method across various experimental settings, including cross-task, cross-domain, cross-training configurations, and out-of-domain scenarios. Additionally, we conducted several experiments to further assess the effectiveness of our method: merging different numbers of tasks (App. C.1 and Fig. 8), comparison with AdaMerging on vision tasks (App. C.2 and Tab. 7), and providing additional results using evolutionary strategies (ES) (App. C.3 and Tab. 8). Lastly, we present comprehensive task-level results in App. C.4.

### 5.1 Cross Task Merging

**Merging NLP Models.** For the NLP domain, we adhere to the experimental setting from [89]. We employ the T5-base and T5-large [59] models and fine-tune both on seven tasks. This setting considers a variety of NLP domains such as question answering, paraphrase identification, sentence completion, and coreference resolution (dataset details in App. D). Tab. 2 shows that using PCB-MERGING to merge fully fine-tuned T5-base and T5-large models leads to an average improvement of 4.3% and 3.5% over 7 tasks, without extra data. With validation datasets, PCB-MERGING improves by 1.8% and 1.8% over other methods for T5-base and T5-large, respectively. Notably, PCB-MERGING without validation outperforms TIES-merging [89] by 5.4% for T5-large. For more detailed results, refer to App. Tab. 9 and 10.

**Merging PEFT Model Adapters.** Following the work of [89], we consider merging parameters used for efficient fine-tuning calculations and employ the $(IA)^3$ [42] method for experimentation. This approach, a form of Parameter-Efficient Fine-Tuning (PEFT), extends the activations of base models with learned vectors. We select T0-3B [66] as the base model and fine-tune $(IA)^3$ models on the training sets of eleven datasets, including sentence completion, natural language inference, coreference resolution, and word sense disambiguation (dataset details in App. D). During fine-tuning of the T0-3B model, we utilize prompt templates from the Public Prompt Pool (P3 [4]) to convert

Table 2: Comparison of different model merging methods across various fine-tuning configurations and modalities, with average performance reported for different tasks.

| Task (→)
Method (↓) | Validation | 7 **NLP** Tasks | | 11 **PEFT** Tasks | 3 **LLM** Tasks | 8 **Vision** Tasks | |
|---|---|---|---|---|---|---|---|
| | | T5-Base | T5-Large | (IA)$^3$ | LLaMa2 | ViT-B/32 | ViT-L/14 |
| Fine-tuned | - | 83.1 | 88.9 | 71.4 | 40.4 | 90.5 | 94.2 |
| Multitask | - | 83.6 | 88.1 | 73.1 | - | 88.9 | 93.5 |
| Averaging[ICML22] [86] | ✗ | 65.3 | 54.7 | 57.9 | 30.3 | 65.8 | 79.6 |
| Task Arithmetic[ICLR23] [28] | ✗ | 53.5 | 73.6 | 59.2 | 30.4 | 60.4 | 83.3 |
| Ties-Merging[NeurIPS23] [89] | ✗ | 69.5 | 71.7 | 64.9 | 34.2 | 72.4 | 86.0 |
| **PCB-MERGING (ours)** | ✗ | **73.8** (+4.3) | **77.1** (+3.5) | **66.1** (+1.2) | **35.1** (+0.9) | **75.9** (+3.5) | **86.9** (+0.9) |
| Fisher Merging[NeurIPS22] [46] | ✓ | 68.3 | 68.7 | 62.2 | - | 68.3 | 82.2 |
| RegMean[ICLR23] [30] | ✓ | 72.7 | 79.8 | 58.0 | - | 71.8 | 83.7 |
| Task Arithmetic[ICLR23] [28] | ✓ | 73.0 | 80.2 | 63.9 | 30.4 | 70.1 | 84.5 |
| Ties-Merging[NeurIPS23] [89] | ✓ | 73.6 | 80.3 | 66.8 | 34.2 | 73.6 | 86.0 |
| **PCB-MERGING (ours)** | ✓ | **75.4** (+1.8) | **82.1** (+1.8) | **68.1** (+1.3) | **35.1** (+0.9) | **76.3** (+2.7) | **87.5** (+1.5) |
| **PCB-MERGING + ES (ours)** | ✓ | **76.7** (+3.1) | **83.2** (+2.9) | **68.8** (+2.0) | **35.3** (+1.1) | **77.0** (+3.4) | **88.1** (+2.1) |

each example in each dataset into a text-to-text format, where each label corresponds to a different string. For experiments with (IA)$^3$, we report the median score across all templates for each dataset. Tab. 2 illustrates that PCB-MERGING achieves an average improvement of 1.2% and 1.3% across 11 tasks compared to the top baseline, both with and without validation set. For further details, please refer to App. Tab. 11.

**Merging LLMs.** In our experiment, we merged three specialized large language models based on the Llama-2-7b architecture [80]—focusing on Chinese language proficiency$^3$, mathematical reasoning [93]$^4$, and code generation [63]$^5$. Each model was assessed using tailored benchmarks: CMMLU [38] for Chinese, GSM8K [10] for math, and HumanEval [6] for code generation (dataset details in App. D). As shown in Tab. 3, PCB-MERGING improved overall performance by an average of 0.8% (no DARE) and 0.6% (with DARE). The most significant performance gain was in code generation, with 3.7% improvement without DARE and 2.5% with DARE [94]. The results indicate that although the DARE preprocessing provided modest improvements, our proposed methodology notably enhanced the overall performance.

Table 3: Comparison of the performance of different methods on 3 datasets after merging LLMs.

| Model | DARE | CMMLU | GSM8K | Human-Eval | Average |
|---|---|---|---|---|---|
| Chinese | - | 38.6 | 2.3 | 13.4 | 18.1 |
| Math | - | 31.2 | 65.6 | 0 | 32.3 |
| Code | - | 33.3 | 0 | 17.1 | 16.8 |
| Averaging
[ICML22] [86] | ✗ | 35.6 | 48.5 | 6.7 | 30.3 |
| | ✓ | 35.6 | 47.8 | 8.5 | 30.7 |
| Task Arithmetic
[ICLR23] [28] | ✗ | 35.4 | 46.1 | 9.8 | 30.4 |
| | ✓ | 35.5 | 46.1 | 10.4 | 30.7 |
| TIES-Merging
[NeurIPS23] [89] | ✗ | 36.5 | 53.4 | 12.8 | 34.3 |
| | ✓ | 36.4 | 53.4 | 14.0 | 34.6 |
| **PCB-MERGING**
**(ours)** | ✗ | 36.4 | 52.3 | 16.5 | 35.1 |
| | ✓ | 36.5 | 52.7 | 16.5 | 35.2 |
| **PCB-MERGING + ES**
**(ours)** | ✗ | 36.4 | 53.1 | 16.5 | **35.3** |
| | ✓ | 36.4 | 53.8 | 16.5 | **35.6** |

**Merging Vision Models.** For image classification tasks, we adopt the experimental setup outlined by Ilharco et al. [27, 28]. We utilize two versions of the CLIP model [58] featuring ViT-B/32 and ViT-L/14 models [12] as visual encoders. Subsequently, we fine-tune the visual encoder on eight tasks sourced from Ilharco et al. [28] and Radford et al. [58], while maintaining the text encoder unchanged. This configuration encompasses diverse classification domains including remote sensing, traffic classification, and satellite imagery recognition (dataset details in App. D). PCB-MERGING performs better than the top baseline by 3.5% and 0.9% for ViT-B/32 and ViT-L/14, respectively, when validation is not utilized. With additional data, these improvements are 2.7% and 1.5%, respectively, and further increase to 3.4% and 2.1% after incorporating evolutionary search. For more detailed findings, please refer to App. Tab. 12, 13 and Fig. 9.

---

$^3$https://huggingface.co/LinkSoul/Chinese-Llama-2-7b
$^4$https://huggingface.co/meta-math/MetaMath-7B-V1.0
$^5$https://huggingface.co/qualis2006/llama-2-7b-int4-python-code-18k

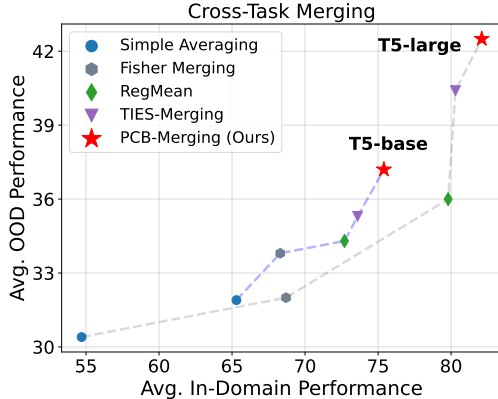
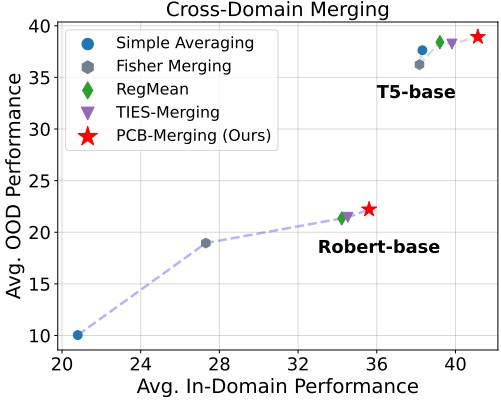

Figure 4: Comparison of average performance on 7 in-domain and 6 held-out datasets after cross-task merging.

Figure 5: Comparison of average performance on 5 in-domain and 5 distribution shift datasets after cross-domain merging.

**Out of Domain Gegeralization.** Following the experimental setup of [89], we also examined the ability of cross-task merged models to better generalize across different domains. We merged the T5-base and T5-large models using the same approach as in the previous experiments, combining them on seven in-domain datasets. Subsequently, we evaluated their performance on six held-out datasets from the T0 mixture [66] to assess out-of-domain generalization. These out-of-domain datasets encompass various tasks, including question answering, word sense disambiguation, and sentence completion (details in App. D). Both in-domain and out-of-domain performance are presented together in Fig. 4. The results show that PCB-MERGING outperforms the strongest baseline for both T5-base and T5-Large models by 1.9% and 2.1%, respectively, indicating superior out-of-domain generalization. For more detailed results, please refer to App. Tab. 14.

## 5.2 Cross Domain Merging

We conducted further experiments to compare the performance of different methods in merging five distinct domain-specific models for emotion classification. Following the methodology of Jin et al. [30], we employed the Roberta-base and T5-base models and utilized a set of preprocessed datasets from Ober et al. [53]. For training individual models, we selected five high-resource datasets, while five low-resource datasets were chosen for evaluating out-of-domain generalization ability. Our analysis reports the average accuracy of in-domain datasets and the average accuracy of out-of-domain datasets using various model merging techniques. In addition, we conducted the experiment with different random seeds and reported the average results across five seeds. Fig. 5 provides a summarized overview of these results. Our findings indicate that PCB-MERGING outperforms the strongest baseline by 1.1% for Roberta-base and 1.3% for T5-base, while improving generalization across domain shifts by 0.8% and 0.7%, respectively. Further details regarding the datasets can be found in App. D and Tab. 16, and additional results are provided in App. C.4 and Tab. 15.

## 5.3 Cross Training Configurations Merging

In this experiment, our main focus was to compare the ability of methods to merge multiple checkpoints of the same task. These checkpoints were generated by employing different training configurations during fine-tuning, which included variations in hyperparameters, augmentation strategies, and dataset partitioning. Following the setup of model soups [86], we fine-tuned RoBERT-base [44] models on four text classification tasks from the GLUE benchmark [82]: MRPC [11], RTE [19], CoLA [84] and SST-2 [73].

Table 4: Comparison of the performance of different methods on 4 datasets after merging multiple checkpoints with various training configurations.

| Method | MRPC | RTE | COLA | SST-2 | 4-task Avg. |
|---|---|---|---|---|---|
| Avg. Individuals | 81.7 | 65.2 | 43.1 | 86.5 | 69.1 |
| Averaging[ICML22] [86] | 79.7 | 59.4 | 37.8 | 87.2 | 66.0 |
| Fisher[NeurIPS22] [46] | 83.3 | 65.4 | 53.4 | 88.6 | 72.7 |
| RegMean[ICLR23] [30] | 81.2 | 66.8 | 48.7 | 88.1 | 71.2 |
| Task Arithmetic[ICLR23] [28] | 81.9 | 68.7 | 42.3 | 87.9 | 69.7 |
| TIES-Merging[NeurIPS23] [89] | 84.2 | 69.3 | 55.7 | 88.9 | 74.5 |
| **PCB-MERGING (ours)** | **85.3** | **70.3** | **58.4** | **89.2** | **75.8** |

We fine-tuned 10 models for each dataset using a random hyperparameter search over learning rate, batch size, and number of epochs (training details in App. F.2). Additionally, we randomly selected training subsets with 1000 examples from the entire training datasets, resulting in each subset having different label distributions. We use the standard metric for each dataset: average of accuracy and $F_1$ score for MRPC, accuracy for RTE, Matthews correlation [47] for CoLA and accuracy for SST-2. We repeated this experiment with different random seeds and reported the average results across five seeds. Tab. 4 presents the corresponding metrics on the validation set, showing consistent performance improvements with PCB-MERGING across all datasets.

## 6   Analysis

### 6.1   Ablation of PCB-MERGING Components

We conducted ablation experiments on various components of our approach to assess their importance. Tab. 5 compares the performance of our method with different components removed, testing ViT-B/32 and T5-base models on the validation set. Removing the *Rescale* step implies using a uniform scale $\lambda = 1$ and computing a disjoint mean as in TIES-Merging [89], ignoring zero values. The table demonstrates the crucial importance of all components for achieving optimal performance. Specifically, the *Drop* component was found to be the most critical, resulting in performance drops of $5.1\%$ for ViT-B/32 and $4.9\%$ for T5-base, respectively. More ablation study details are provided in App. B.1 and Tab. 6.

Table 5: Ablation study on individual components of PCB-MERGING.

| Task($\rightarrow$) | Vision | NLP |
|---|---|---|
| Method($\downarrow$) | ViT-B/32 | T5-base |
| w/o Intra-Balance | 74.4 | 73.7 |
| w/o Inter-Balance | 74.8 | 73.9 |
| w/o Drop | 71.2 | 70.5 |
| w/o Rescale | 73.8 | 72.9 |
| PCB-MERGING | **76.3** | **75.4** |

### 6.2   Effect of Hyper-Parameters on the Performance.

We examined the impact of hyper-parameters $\lambda$ and $r$ on the performance when merging multiple NLP tasks, as discussed in Section 5.1. Initially, we illustrate the performance of various models across different values of $\lambda$ while keeping $r = 0.1$. Our method is compared against the state-of-the-art baseline method, TIES-Merging. From Fig. 6, We can observe that our approach demonstrates a higher performance ceiling within the suitable range of 1.4 to 1.8. As $\lambda$ increases, the performance initially decreases and then saturates. Additionally, we provide a performance analysis for different ratios $r$. We conduct a grid search for $\lambda$ to determine its optimal performance for each ratio. Notably, for $r < 0.3$, our method consistently showcases significant improvements. This underscores the importance of the information filtered out by our parameter competition balancing approach in the merging process. More analysis about hyper-parameters are shown in App. B.2 and Fig. 7.

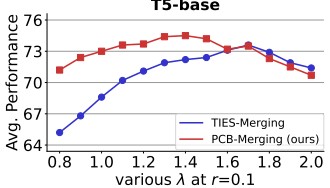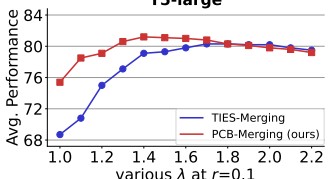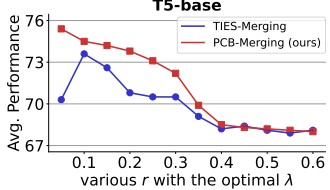

Figure 6: Performance with various hyperparameters $\lambda$ and $r$.

### 6.3   Limitation and Future Work

While our approach provides valuable insights into model merging, several limitations should be noted: (1) PCB-MERGING, like previous methods, relies on identical model architectures and shared initializations, constraining its applicability across various model types. (2) Limited theoretical understanding: model merging effectiveness may be influenced by task independence [34] and weight disentanglement [55, 54], warranting further exploration. (3) Our approach does not effectively address parameter redundancy, still relying on drop operations to mitigate interference and improve performance. (4) Task vector magnitudes may not always effectively represent parameter importance, necessitating further exploration for more efficient methods.

# 7   Conclusions

In summary, we introduce PCB-MERGING to tackle challenges in model merging by incorporating parameter competition balancing to rescale task vectors at the parameter level. Our method enhances model merging performance without requiring additional training, leading to improved stability and effectiveness across various scenarios. We demonstrate significant advancements in cross-task merging, cross-domain merging, different training configurations, and out-of-domain generalization, highlighting its potential impact in practical applications.

# Acknowledgements

We thank all the reviewers for their valuable feedback on this paper. This work was supported in part by National Science Foundation of China (62476070, 62376074, 12204130), Shenzhen College Stability Support Plan (GXWD20231128103232001) and Department of Science and Technology of Guangdong (2024A1515011540), the Shenzhen Science and Technology Program (Grants:JSGGKQTD20221101115655027, RKX20231110090859012, SGDX20230116091244004), Shenzhen Start-Up Research Funds (Grant No.HA11409065), and the Fundamental Research Funds for the Central Universities (Grant No. HIT.OCEF.2024047).

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

# Appendix for PCB-Merging

## A Novelty and Contribution

Our research aims to unlock the full potential of task vector-based approaches by adjusting coefficients at the parameter level through a balancing mechanism that addresses parameter competition across different tasks. We re-examine existing model merging methods and highlight the critical role of parameter competition awareness. To clearly demonstrate the innovation of our method, we conduct a comparative analysis with existing state-of-the-art baseline methods.

**Comparison with TIES-Merging** Both the TIES-Merging [89] and our approach address parameter competition or interference through self-awareness and cross-awareness. However, there are several key differences:

1. When performing *Drop / Trim* to reduce redundancy, we consider both intra-competition and inter-competition, whereas TIES-Merging primarily considers parameter magnitude.

2. In terms of cross-awareness, TIES-Merging only considers the direction of parameters across different tasks, neglecting parameter weights. Our method more accurately measures the similarity of task vectors to assess conflict levels. We conducted ablation experiments to demonstrate the effectiveness of inter-balancing, as shown in App. B.1 and Tab. 6.

3. Our approach modulates the coefficient of each parameter, while TIES-Merging uses a uniform scale for all tasks and parameters. Ablation experiments in the Analysis section validate the superiority of our method, as shown in Section 6.1 and Tab. 5.

**Comparison with AdaMerging** Although AdaMerging [90] has achieved significant performance improvements in image classification, it has several drawbacks:

1. This method requires unsupervised test samples, which is often impractical.

2. The use of Shannon entropy to train the adaptive weights limits the method to classification tasks.

3. AdaMerging requires unsupervised training with the availability of (unlabeled) test samples, which is a different setup than generalizing to an entirely unseen test set.

In contrast, our proposed PCB-Merging retains the efficiency and lightwight nature as most previous merging methods. Additionally, we conducted experiments on image classification tasks to compare the two methods, as shown in App. C.2 and Tab. 7.

**Comparison with Fisher Merging and RegMean** The same as Fisher Merging [46] and Reg-Mean [30], our PCB-Merging method also introduces additional matrices to adjust parameter coefficients, but there are two key differences:

1. Fisher Merging and RegMean consider only self-awareness or cross-awareness, respectively. In contrast, our method accounts for various scenarios of parameter competition.

2. Both Fisher Merging and RegMean require additional gradient-based computations to obtain the Fisher Information Matrix or Inner Product Matrix, which demand more GPU resources. Our method, however, is based on task vectors, making it easier and lightwight to implement.

**Comparison with DARE** Both DARE [94] and PCB-Merging drop and rescale task vectors for model merging, but there are significant differences:

1. DARE randomly drops parameters according to a drop rate $p$, while we consider parameter competition.

2. DARE rescales the remaining parameters by a uniform factor of $1/(1-p)$, whereas we compute a specific coefficient for each task and each parameter.

3. DARE is mainly used in LLM model merging to maintain the original fine-tuned performance. In contrast, we find that dropping parameters can further enhance performance beyond the fine-tuned model with a suitable scale and intra-balancing.

**Comparison with Lorahub** Lorahub [25] aims to establish a strategic framework for composing LoRA modules trained on diverse tasks to achieve adaptable performance on new tasks. This framework utilizes an evolution algorithm (CMA-ES [21]) to search for the coefficients of each LoRA module, as introduced in Section 3.3. However, this search-based approach is time-consuming and can only be applied at the task level, leading to limited performance. Moreover, LoRA lacks self-awareness and considers only competition between different tasks.

**Comparison with Task Arithmetic and PEM Compositon** Both Task Arithmetic [28] and PEM Composition [96] methods primarily focus on exploring potential applications of task vectors, including distribution generalization, unlearning, and domain transfer. However, they do not address parameter competition or balance the coefficients of different tasks or parameters, which limits their performance.

---

**Algorithm 1** PCB-Merging Procedure.
___

**Input:** Fine-tuned models $\{\theta_i\}_{i=1}^n$, Initialization $\theta_{\text{pre}}$, mask ratio $r$ and coefficient $\lambda$.
**Output:** Merged Model $\theta_m$
▷ Create task vectors.
$\{\tau_i\}_{i=1}^n = \{\theta_i\}_{i=1}^n - \theta_{\text{pre}}$
**for** $i$ **in** $1, ..., n$ **do**
   ▷ Step 1: Intra-Balancing.
   $\beta_{intra,i} = \text{Softmax}(N * \text{Norm}(\tau_i \odot \tau_i))$
   ▷ Step 2: Inter-Balancing.
   $\beta_{inter,i} = \sum_{j=1}^n \text{Softmax}(\tau_i \odot \tau_j)$
   ▷ Step 3: Drop low-scoring parameters.
   $\beta_i = \beta_{intra,i} \odot \beta_{inter,i}$
   $m_i = \beta_i \geq \text{sorted}(\beta_i)[(1-r) \times D]$
   $\hat{\beta}_i = m_i \odot \beta_i$
**end**
▷ Step 4: Rescale task vectors.
$\tau_m = \sum_{i=1}^n (\hat{\beta}_i \odot \tau_i) / \sum_{i=1}^n \hat{\beta}_i$
▷ Obtain merged checkpoint
$\theta_m \leftarrow \theta_{\text{init}} + \lambda * \tau_m$
**return** $\theta_m$

---

# B  Additional Analysis

## B.1  Additional Ablation Studies

We present additional ablation experiments on PCB-MERGING, as shown in Tab. 6. In addition to the four main steps discussed in Section 6.1 (Intra-Balancing, Inter-Balancing, Drop, and Rescale), we also tested other influencing factors:

1. Activation functions: We replaced the softmax activation function with common alternatives like sigmoid, ReLU, and tanh. The results show minimal performance loss with different activation functions, except for ReLU in intra-balancing. This is because these activation functions can represent complex nonlinear relationships to balance the values of parameters.

2. Without regulator N: We removed the regulator N in intra-balancing, which controls intra-competition according to the number of models being merged.

3. Inter-balancing with only sign: We computed inter-balancing using only the sign $(-1, 1)$ instead of the actual values, where the sign represents a direction in the $D$-dimensional parameter space relative to initialization. This experiment aims to compare with TIES-Merging, which addresses sign conflicts.

4. Element-wise multiplication vs. Addition: We combined intra-balancing and inter-balancing using addition instead of multiplication. This resulted in a performance loss of 4.1% and 3.9% on the ViT-B/32 and T5-base models, respectively.

In summary, these ablation experiments demonstrate the functionality and impact of each component in our method.

Table 6: More extensive ablation studies on PCB-MERGING

| Ablation (→) | activation in intra-balancing | | | activation in inter-balancing | | | without | inter-balancing | replace multiplication | PCB |
| Model (↓) | sigmoid | relu | tanh | sigmoid | relu | tanh | regulator N | with only sign | by adding | Merging |
|---|---|---|---|---|---|---|---|---|---|---|
| ViT-B/32 | 76.1 | 74.9 | 76.1 | 76.2 | 76.1 | 76.4 | 74.7 | 75.7 | 72.2 | **76.3** |
| T5-base | 75.3 | 72.8 | 75.2 | 75.3 | 75.2 | 75.4 | 74.1 | 74.5 | 71.5 | **75.4** |

## B.2  Additional Hyper-parameters Analysis

In this section, we present additional experimental results regarding hyper-parameters, observing similar phenomena and conclusions as those in Section 6.2. We explored the effects of $\lambda$ and $r$ on

the performance of merging multiple NLP tasks, as discussed in Section 5.1. First, we show the performance of various models for different values of $\lambda$, keeping $r = 0.2$. Our method is compared to the state-of-the-art baseline, TIES-Merging. As shown in Fig. 7, our approach achieves a higher performance ceiling within the optimal range of 0.8 to 1.6. As $\lambda$ increases, the performance initially decreases and then levels off.

Furthermore, we provide a performance analysis for different values of $r$ with T5-large. We conducted a grid search for $\lambda$ to find its optimal performance for each ratio. Significantly, for $r < 0.4$, our method consistently shows substantial improvements. This highlights the importance of the information filtered by our parameter competition balancing approach in the merging process.

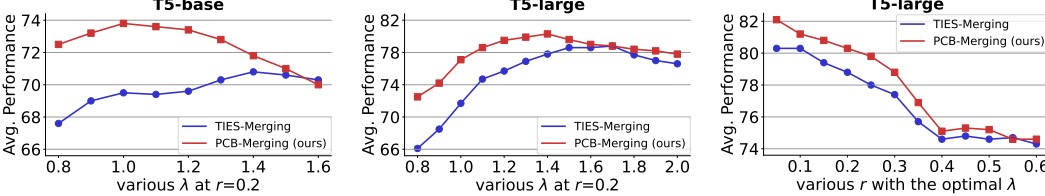

Figure 7: Performance with various hyperparameters $\lambda$ and $r$.

## C Additional Results

### C.1 Merging Different Number of Tasks

We evaluated the performance of the merged model on in-domain tasks and analyzed how it varies with the number of tasks being merged. In Fig. 8, we normalized each task's accuracy to its fine-tuned model's performance and reported the average normalized accuracy for in-domain tasks with T5-base model. We compared our method against the strongest baseline, TIES-Merging [89], and simple averaging [86]. Each data point represents the merging of a subset of tasks, with the solid line indicating the average performance across multiple subsets. We observed that as the number of merged tasks increases, the performance of all methods de-

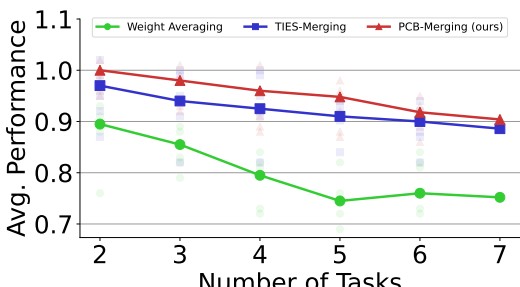

Figure 8: Average normalized performance when merging a different number of tasks.

clines, suggesting that more tasks lead to increased parameter competition. Additionally, TIES-Merging's performance drops faster than PCB-Merging, indicating that our PCB-Merging method is more effective in balancing parameter competition.

### C.2 Compare with Adamerging

We conducted cross-task merging experiments on image classification tasks to compare our method with AdaMerging [90]. AdaMerging employs unsupervised training to learn merging coefficients for each task vector in Task Arithmetic using unlabeled test datasets. Additionally, Layer-wise AdaMerging learns coefficients for each layer of each task vector.

Table 7: Compare the performance of different merging methods after applying unsupervised training with AdaMerging.

| Model | Coefficient | AdaMerge | Ada + TIES | Ada + PCB |
|-------|-------------|----------|------------|-----------|
| ViT-B/32 | Task-wise | 71.8 | 74.9 | **77.1** |
|          | Layer-wise | 80.1 | 81.1 | **81.7** |
| ViT-L/14 | Task-wise | 85.6 | 86.8 | **88.2** |
|          | Layer-wise | 90.8 | 91.0 | **91.3** |

AdaMerging can be further improved by applying strategies from TIES-Merging to modify task vectors or using PCB-Matrix to adjust the task vectors. As shown in Tab. 7, our method enhances AdaMerging, resulting in performance improvements of 2.2% and 1.4% on the ViT-B/32 and ViT-L/14 models, respectively.

## C.3 Compare with TIES-Merging using Evolutionary Strategy

To validate the effectiveness of the evolutionary strategy (ES) proposed in Section 3.3, we applied ES to intelligently search for coefficients of different tasks in other baseline methods. The results are shown in Tab. 8. Notably, after applying ES, TIES-Merging showed significant improvement. We also compared TIES-Merging with ES against our approach with ES. The results demonstrate the effectiveness of PCB-MERGING, particularly with a 2.2% performance gain on the T5-large model.

Table 8: Comparing the performance of different methods with evolutionary strategies (ES) after cross-task merging.

| Task (→) | 7 **NLP** Tasks | | 11 **PEFT** Tasks | 3 **LLM** Tasks | 8 **Vision** Tasks | |
|---|---|---|---|---|---|---|
| Method (↓) | T5-Base | T5-Large | (IA)$^3$ | LLaMa2 | ViT-B/32 | ViT-L/14 |
| Ties-Merging | 73.6 | 80.3 | 66.8 | 34.2 | 73.6 | 86.0 |
| **PCB-MERGING (ours)** | **75.4** (+1.8) | **82.1** (+1.8) | **68.1** (+1.3) | **35.1** (+0.9) | **76.4** (+2.8) | **87.5** (+1.5) |
| Ties-Merging + ES | 74.8 | 81.0 | 67.6 | 34.3 | 74.9 | 86.8 |
| **PCB-MERGING + ES (ours)** | **76.7** (+1.9) | **83.2** (+2.2) | **68.8** (+1.2) | **35.3** (+1.0) | **77.0** (+2.1) | **88.1** (+1.6) |

## C.4 Comprehensive Task-Level Results

We provide the task level for all the cross-task merging experiments in the main Tab. 2. Tab. 9, 10, 11, 12, and 13 provide the task level results T5-Base, T5-Large [59], IA3 [42], ViT-B/32, and ViT-L/14 [12] respectively. The task level results of the out-of-domain experiments for T5-Base and T5-Large can be found in Tab. 14.

Table 9: Test set performance when merging T5-base models on seven NLP tasks. Please refer to Section 5.1 for experimental details.

| Task(→) | Validation | Average | Test Set Performance | | | | | | |
|---|---|---|---|---|---|---|---|---|---|
| Method(↓) | | | paws | qasc | quartz | story_cloze | wiki_qa | winogrande | wsc |
| **Zeroshot** | - | 53.5 | 49.9 | 35.8 | 53.3 | 48.1 | 76.2 | 50 | 61.1 |
| **Fine-tuned** | - | 83.1 | 94.6 | 98.4 | 81.1 | 84.9 | 95.8 | 64.5 | 62.5 |
| **Multitask** | - | 83.6 | 94 | 97.9 | 82.5 | 86.7 | 95 | 64.1 | 65.3 |
| **Averaging**[ICML22] [86] | ✗ | 65.3 | 67.4 | 83.4 | 60.8 | 50.3 | 93.2 | 51.7 | 50.0 |
| **Task Arithmetic**[ICLR23] [28] | ✗ | 53.5 | 50.6 | 22.4 | 55.0 | 63.6 | 79.2 | 53.9 | 50.0 |
| **Ties-Merging**[NeurIPS23] [89] | ✗ | 69.5 | 76.1 | 79.5 | 68.5 | 65.6 | 86.3 | 56.2 | 54.2 |
| **PCB-MERGING (ours)** | ✗ | 73.8 | 77.1 | 91.5 | **68.5** | 75.8 | 88.2 | **61.1** | 54.2 |
| **Fisher Merging**[NeurIPS22] [46] | ✓ | 68.3 | 66.7 | 85.6 | 63.5 | 57.1 | 90.1 | 54.2 | 60.8 |
| **RegMean**[ICLR23] [30] | ✓ | 72.7 | 77.2 | **93.8** | 63.6 | 64.6 | 90.4 | 58.4 | 60.7 |
| **Task Arithmetic**[ICLR23] [28] | ✓ | 73.0 | 69.6 | 91.5 | 67.3 | 76.1 | 91.3 | 58.3 | 56.9 |
| **Ties-Merging**[NeurIPS23] [89] | ✓ | 73.6 | **82.2** | 84.8 | 66.1 | 73.5 | 87.0 | 60.2 | 61.1 |
| **PCB-MERGING (ours)** | ✓ | **75.4** | 79.0 | 93.2 | 65.8 | **76.1** | **89.9** | 59.8 | **63.9** |

Table 10: Test set performance when merging T5-large models on seven NLP tasks. Please refer to Section 5.1 for experimental details.

| Task(→) | Validation | Average | Test Set Performance | | | | | | |
|---|---|---|---|---|---|---|---|---|---|
| Method(↓) | | | paws | qasc | quartz | story_cloze | wiki_qa | winogrande | wsc |
| **Zeroshot** | - | 53.1 | 58.2 | 54.2 | 54.1 | 54.3 | 70.9 | 49.2 | 63.9 |
| **Fine-tuned** | - | 88.9 | 94.5 | 98.3 | 88.5 | 91.4 | 96.2 | 74.5 | 79.2 |
| **Multitask** | - | 88.1 | 94.2 | 98.5 | 89.3 | 92 | 95.4 | 73.5 | 73.6 |
| **Averaging**[ICML22] [86] | ✗ | 54.7 | 57.2 | 26.4 | 71.4 | 54.8 | 86.6 | 50.2 | 36.1 |
| **Task Arithmetic**[ICLR23] [28] | ✗ | 73.6 | 69.7 | 83.6 | 58.3 | 77.4 | 94.4 | 59.3 | 72.2 |
| **Ties-Merging**[NeurIPS23] [89] | ✗ | 71.7 | 71.2 | 97.1 | 74.2 | 74.9 | 73.3 | 62.9 | 48.6 |
| **PCB-MERGING (ours)** | ✗ | 77.1 | 78.1 | 98 | **75.4** | 77.7 | 89.1 | 64.6 | 56.9 |
| **Fisher Merging**[NeurIPS22] [46] | ✓ | 68.7 | 68.4 | 83 | 65.5 | 62.4 | 94.1 | 58.2 | 49.2 |
| **RegMean**[ICLR23] [30] | ✓ | 79.8 | **83.9** | 97.2 | 73.2 | 82.6 | 94.1 | 63.2 | 64.4 |
| **Task Arithmetic**[ICLR23] [28] | ✓ | 80.2 | 77.6 | 96.6 | 75.1 | **85.6** | 93.8 | 61.8 | 70.8 |
| **Ties-Merging**[NeurIPS23] [89] | ✓ | 80.3 | 78.2 | 97.5 | 72.8 | 83.7 | **94.5** | 64.5 | 70.8 |
| **PCB-MERGING (ours)** | ✓ | **82.1** | 82.0 | **98.4** | 72.2 | **85.6** | 94.0 | **67.5** | **75.0** |

Table 11: Test set performance when merging (IA)$^3$ models on eleven tasks. Please refer to Section 5.1 for experimental details.

| Task(→) Method(↓) | Validation | Average | Natural Language Inference | | | | | Sentence Completion | | | Co-reference | | WSD |
|---|---|---|---|---|---|---|---|---|---|---|---|---|---|
| | | | RTE | CB | ANLI1 | ANLI2 | ANLI3 | COPA | Hella. | Story. | WSC | Wino. | WiC |
| Zeroshot | - | 53.1 | 58.2 | 54.2 | 35.5 | 34.4 | 34.4 | 75.0 | 39.2 | 86.5 | 63.9 | 51.2 | 51.9 |
| Fine-Tuned | - | 71.4 | 82.7 | 95.8 | 70.4 | 46.5 | 53.0 | 85.3 | 44.4 | 95.0 | 65.3 | 75.1 | 71.7 |
| Averaging[ICML22] [86] | - | 57.9 | 81.2 | 58.3 | 43.3 | 39.1 | 40.0 | 80.9 | 40.1 | 92.4 | 52.8 | 53.8 | 55.0 |
| Task Arithmetic[ICLR23] [28] | ✗ | 59.2 | 76.5 | 79.2 | 59.8 | 47.5 | 48.2 | 66.2 | 31.4 | 81.5 | 51.4 | 57.7 | 51.6 |
| TIES-Merging[NeurIPS23] [89] | ✗ | 64.9 | 81.2 | 87.5 | 58.1 | 46.5 | 47.4 | 80.2 | 42.6 | 91.1 | 58.3 | 60.8 | 59.9 |
| PCB-MERGING (ours) | ✗ | 66.1 | 85.9 | 83.3 | 64.2 | 47.8 | 45.9 | 82.4 | 42.7 | 91.2 | 63.9 | 61.9 | 57.1 |
| Fisher Merging[NeurIPS22] [46] | ✓ | 62.2 | 83.3 | 83.3 | 45.9 | 41.0 | 42.2 | 83.1 | 42.2 | 94.1 | 58.3 | 56.7 | 54.2 |
| RegMean[ICLR23] [30] | ✓ | 58 | 81.2 | 58.3 | 43.3 | 39.2 | 40.2 | 80.9 | 40.1 | 92.5 | 53.5 | 53.8 | 55 |
| Task Arithmetic[ICLR23] [28] | ✓ | 63.9 | 74.1 | 83.3 | 60.8 | 49.4 | 50.0 | 87.5 | 41.5 | 95.3 | 49.3 | 62.8 | 49.1 |
| TIES-Merging[NeurIPS23] [89] | ✓ | 66.8 | 78.6 | 87.5 | 66.6 | 51.3 | 51.5 | 81.7 | 43.2 | 90.9 | 57.6 | 67.0 | 58.4 |
| PCB-MERGING (ours) | ✓ | 68.1 | 80.0 | 83.3 | 67.1 | 51.1 | 49.6 | 88.3 | 42.7 | 92.8 | 61.8 | 67.6 | 64.7 |

Table 12: Test set performance when merging ViT-B/32 models on 8 vision tasks. Please refer to Section 5.1 for experimental details.

| Task(→) Method(↓) | Validation | Average | Test Set Performance | | | | | | | |
|---|---|---|---|---|---|---|---|---|---|---|
| | | | SUN397 | Cars | RESISC45 | EuroSAT | SVHN | GTSRB | MNIST | DTD |
| Individual | - | 90.5 | 75.3 | 77.7 | 96.1 | 99.7 | 97.5 | 98.7 | 99.7 | 79.4 |
| Multitask | - | 88.9 | 74.4 | 77.9 | 98.2 | 98.9 | 99.5 | 93.9 | 72.9 | 95.8 |
| Averaging[ICML22] [86] | ✗ | 65.8 | 65.3 | 63.4 | 71.4 | 71.7 | 64.2 | 52.8 | 87.5 | 50.1 |
| Task Arithmetic[ICLR23] [28] | ✗ | 60.4 | 36.7 | 41 | 53.8 | 64.4 | 80.6 | 66 | 98.1 | 42.5 |
| Ties-Merging[NeurIPS23] [89] | ✗ | 72.4 | 59.8 | 58.6 | 70.7 | 79.7 | 86.2 | 72.1 | 98.3 | 54.2 |
| PCB-MERGING (ours) | ✗ | 75.9 | 65.8 | 64.4 | 78.1 | 81.1 | 84.9 | 77.1 | 98.0 | 58.4 |
| Fisher Merging[NeurIPS22] [46] | ✓ | 68.3 | 68.6 | 69.2 | 70.7 | 66.4 | 72.9 | 51.1 | 87.9 | 59.9 |
| RegMean[ICLR23] [30] | ✓ | 71.8 | 65.3 | 63.5 | 75.6 | 78.6 | 78.1 | 67.4 | 93.7 | 52 |
| Task Arithmetic[ICLR23] [28] | ✓ | 70.1 | 63.8 | 62.1 | 72 | 77.6 | 74.4 | 65.1 | 94 | 52.2 |
| Ties-Merging[NeurIPS23] [89] | ✓ | 73.6 | 64.8 | 62.9 | 74.3 | 78.9 | 83.1 | 71.4 | 97.6 | 56.2 |
| PCB-MERGING (ours) | ✓ | 76.3 | 66.7 | 65.5 | 78.5 | 79.3 | 86.4 | 77.1 | 98.2 | 59.1 |

Table 13: Test set performance when merging ViT-L/14 models on 8 vision tasks. Please refer to Section 5.1 for experimental details.

| Task(→) Method(↓) | Validation | Average | Test Set Performance | | | | | | | |
|---|---|---|---|---|---|---|---|---|---|---|
| | | | SUN397 | Cars | RESISC45 | EuroSAT | SVHN | GTSRB | MNIST | DTD |
| Fine-tuned | - | 94.2 | 82.3 | 92.4 | 97.4 | 100 | 98.1 | 99.2 | 99.7 | 84.1 |
| Multitask | - | 93.5 | 90.6 | 84.4 | 99.2 | 99.1 | 99.6 | 96.3 | 80.8 | 97.6 |
| Averaging[ICML22] [86] | ✗ | 79.6 | 72.1 | 81.6 | 82.6 | 91.9 | 78.2 | 70.7 | 97.1 | 62.8 |
| Task Arithmetic[ICLR23] [28] | ✗ | 83.3 | 72.5 | 79.2 | 84.5 | 90.6 | 89.2 | 86.5 | 99.1 | 64.3 |
| Ties-Merging[NeurIPS23] [89] | ✗ | 86 | 76.5 | 85 | 89.3 | 95.7 | 90.3 | 83.3 | 99 | 68.8 |
| PCB-MERGING (ours) | ✗ | 86.9 | 75.8 | 86 | 89.2 | 96 | 88 | 90.9 | 99.1 | 70 |
| Fisher Merging[NeurIPS22] [46] | ✓ | 82.2 | 69.2 | 88.6 | 87.5 | 93.5 | 80.6 | 74.8 | 93.3 | 70 |
| RegMean[ICLR23] [30] | ✓ | 83.7 | 73.3 | 81.8 | 86.1 | 97 | 88 | 84.2 | 98.5 | 60.8 |
| Task Arithmetic[ICLR23] [28] | ✓ | 84.5 | 74.1 | 82.1 | 86.7 | 93.8 | 87.9 | 86.8 | 98.9 | 65.6 |
| Ties-Merging[NeurIPS23] [89] | ✓ | 86 | 76.5 | 85 | 89.4 | 95.9 | 90.3 | 83.3 | 99 | 68.8 |
| PCB-MERGING (ours) | ✓ | 87.5 | 76.8 | 86.2 | 89.4 | 96.5 | 88.3 | 91 | 98.6 | 73.6 |

Additionally, we present the results of merging vision tasks using radar charts for a more intuitive comparison of performance across each task, as shown in Fig. 9. The previous baseline methods show unstable performance, with poor results in some tasks. In contrast, our method is more robust, achieving near-best performance across all tasks.

We also present task-level results of cross-domain merging experiments, as introduced in Section 5.2. Firstly, we fine-tuned five distinct domain-specific models for Emotion Classification and then employed different model merging methods to obtain a single model. For models with an encoder-only architecture, we used the same shared classification head initialization during merging. We tested the performance of the merged model on the original five domains and its generalization on

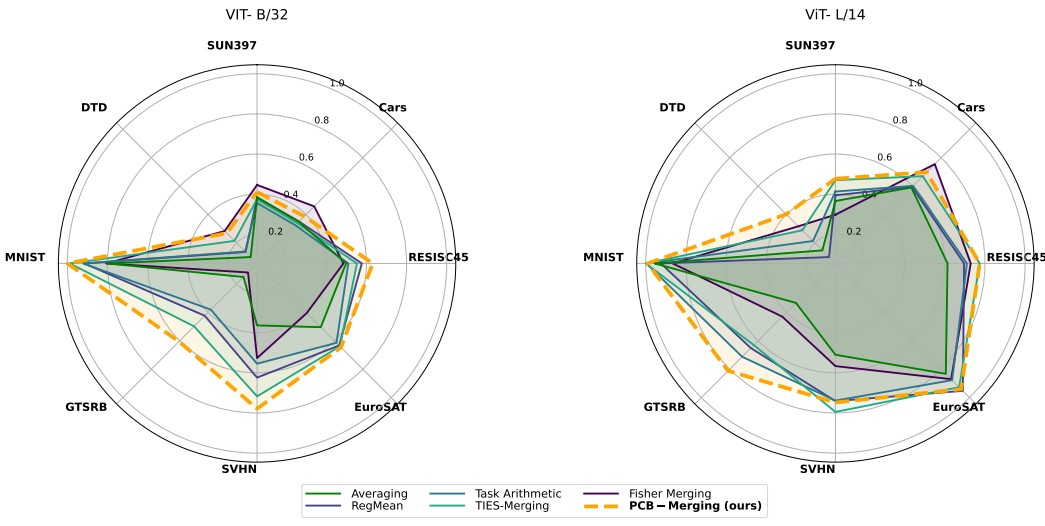

Figure 9: Test set performance when merging ViT-B/32 and ViT-L/14 models on eight image classification tasks.

Table 14: Out-of-distribution performance across six held-out tasks after merging the checkpoints of T5-base and T5-large models from seven NLP tasks. Please refer to Section 5.1 for experimental details.

| Task(→)  Method(↓) | model | Average | Question Answering | | | WSD | Sentence Completion | |
|---|---|---|---|---|---|---|---|---|
| | | | cosmos_qa | social_iqa | quail | wic | copa | h-swag |
| Pretrained | | 31.1 | 21.9 | 18.8 | 24.1 | 65.6 | 43.8 | 12.5 |
| Averaging[ICML22] [86] | | 31.7 | 21.9 | 21.9 | 24.6 | **68.8** | 37.5 | 15.6 |
| Fisher Merging[NeurIPS22] [46] | T5-base | 33.8 | 15.6 | 21.9 | 24.9 | 65.6 | 53.1 | 21.9 |
| Task Arithmetic[ICLR23] [28] | | 31.9 | 15.6 | **31.2** | 25.7 | 28.1 | **68.8** | 21.9 |
| RegMean[ICLR23] [30] | | 34.3 | 23.1 | 28.1 | 24.9 | 48.4 | 62.5 | 18.8 |
| TIES-Merging[NeurIPS23] [89] | | 35.3 | 21.9 | 25 | 25.7 | 50 | 65.6 | 23.8 |
| PCB-MERGING (ours) | | **37.2** | **23.6** | 29.2 | **26.6** | 51.9 | 67.1 | **24.8** |
| Pretrained | | 27.6 | 21.9 | 21.9 | 24.9 | 28.1 | 56.2 | 12.5 |
| Averaging[ICML22] [86] | | 30.4 | 31.2 | 25 | 26.3 | 31.2 | 59.4 | 9.4 |
| Fisher Merging[NeurIPS22] [46] | T5-large | 32 | 34.4 | 25 | 26.1 | 40.6 | 56.2 | 9.4 |
| Task Arithmetic[ICLR23] [28] | | 33.3 | 21.9 | 34.4 | 24.6 | 40.6 | 59.4 | 18.8 |
| RegMean[ICLR23] [30] | | 36 | **34.4** | 28.1 | 25.3 | **62.5** | 50 | 15.6 |
| TIES-Merging[NeurIPS23] [89] | | 40.4 | 31.2 | 43.8 | 26.6 | 59.4 | 59.4 | 21.9 |
| PCB-MERGING (ours) | | **42.5** | 33.6 | **45.8** | **29.6** | 62.2 | **59.2** | **24.6** |

unseen datasets from five other domains. For more dataset details, please refer to App. D. To ensure the reliability of the results, we fine-tuned the models five times with different random seeds and reported the average performance for these runs, as shown in Tab. 15.

Table 15: In domain and Out of domain performance when merging Roberta-base models on 5 emotion datasets. Please refer to Section 5.2 for experimental details.

| Dataset(→)  Method(↓) | In Domain | | | | | | Out of Domain | | | | | |
|---|---|---|---|---|---|---|---|---|---|---|---|---|
| | Average | Dialy. | Crowd. | TEC | Tales | ISEAR | Average | Emoint | SSEC | Elect. | Ground. | Affec. |
| Fine-Tuned | 51.38 | 49.3 | 28.9 | 56.4 | 49.2 | 73.1 | | | | - | | |
| Averaging[ICML22] [86] | 23.2 | 29.9 | 16.6 | 17.0 | 25.2 | 27.1 | 11.6 | 27.8 | 5.2 | 6.5 | 14.0 | 4.3 |
| Fisher Merging[NeurIPS22] [46] | 26.1 | 29.8 | **25.9** | 19.5 | 26.2 | 29.0 | 16.2 | 32.7 | 10.7 | 12.0 | 14.8 | 10.9 |
| RegMean[ICLR23] [30] | 34.2 | **33.1** | 20.7 | 34.1 | 35.0 | 48.3 | 21.3 | 43. | 15.4 | **13.7** | **20. 0** | 14.6 |
| TIES-Merging[NeurIPS23] [89] | 34.5 | 32.2 | 20.6 | 35.5 | 35.1 | 49.3 | 21.5 | 43.4 | 16.1 | 13.3 | 19.7 | 15.0 |
| PCB-MERGING (ours) | **35.6** | 32.1 | 21.2 | **37.4** | **36.0** | **51.2** | **22.2** | **44.2** | **17.5** | 13.5 | 19.7 | **16.1** |

# D Dataset details

This section provides a detailed dataset description.

**Merging NLP Tasks**   Following TIES-Merging [89], we choose seven datasets for merging NLP models: question answering (QASC [32], WikiQA [91], and QuaRTz [78]), paraphrase identification (PAWS [97]), sentence completion (Story Cloze [70]), and coreference resolution (Winogrande [65] and WSC [37]).

**Merging PEFT Models**   Following TIES-Merging [89], we use eleven datasets including sentence completion (COPA [61], H-SWAG [95], and Story Cloze [70] datasets), natural language inference (ANLI [52], CB [45], and RTE [19]), coreference resolution (WSC [37] and Winogrande [65]), and word sense disambiguation (WiC [56]).

**Merging Vision Tasks**   Following Task Arithmetic [28], we study multi-task model merging on eight image classification datasets below. Stanford Cars [35] is a car classification dataset consisting of 196 classes of cars. DTD [9] is a texture classification dataset comprising 47 classes. EuroSAT [22] comprises 10 classes of geo-referenced satellite images. GTSRB [74] includes 43 classes of traffic signs. MNIST [36] features grayscale images of handwritten digits across 10 classes. RESISC45 [7] encompasses 45 classes of remote sensing image scenes. SUN397 [87] consists of 397 classes of scene images. Lastly, SVHN [51] encompasses 10 classes of real-world digital classification images.

**Merging LLMs**

- **CMMLU** [38] is a comprehensive Chinese evaluation benchmark specifically designed to assess language models' knowledge and reasoning abilities in a Chinese context. It covers 67 topics ranging from basic subjects to advanced professional levels.
- **GSM8K** [10] is a collection of 8.5K high-quality, linguistically varied math word problems from grade school, crafted by skilled human authors. The solutions predominantly require executing a series of basic arithmetic operations ($+, -, \times, \div$) to derive the final answer.
- **HumanEval** [6] is a dataset for evaluating code generation ability, containing 164 manually crafted programming problems covering aspects such as language understanding, reasoning, algorithms, and simple mathematics.

Table 16: Statistics of in domain and out-of-domain emotion classification datasets.

|  | Train | Dev | Test |
|---|---|---|---|
| *In-domain* |  |  |  |
| DialyDialog | 72,085 | 10,298 | 20,596 |
| CrowdFlower | 27,818 | 3,974 | 7,948 |
| TEC | 14,735 | 2,105 | 4,211 |
| Tales-Emotion | 10,339 | 1,477 | 2,955 |
| ISEAR | 5,366 | 766 | 1,534 |
| *Out-of-domain* |  |  |  |
| Emoint |  |  | 7,102 |
| SSEC |  |  | 4,868 |
| ElectoralTweets |  |  | 4,056 |
| GroundedEmotions |  |  | 2,585 |
| AffectiveText |  |  | 1,250 |

**Out of Domain Generalilzation**   The average performance is reported over the following tasks and datasets: Cosmos QA [26], Social IQA [67], and QuAIL [62] for question answering; WiC [56] for word sense disambiguation; and COPA [61], and H-SWAG [95] for sentence completion.

**Cross-Domain Merging**   In order to investigate the performance of the sentiment classification task, following RegMean [30], we selected a diverse and challenging set of datasets. Among them, DailyDialogs [41], CrowdFlower, TEC [49], Tales-Emotion [2], and ISEAR [68] is utilized to train domain-specific model. For acessing OOD generalization performance, we use Emoint [48], SSEC [69], ElectoralTweets [50], GroundedEmotions [43], and AffectiveText [76]. For OOD evaluation, we focus exclusively on the fundamental emotions: anger, disgust, fear, joy, sadness, and surprise. A detailed overview of the datasets and statistics is provided in Tab. 16.

**Cross-Training Configurations Merging**   We study four GLUE benchmark text classification datasets [82]. (1) MRPC [11]: Sentence pairs labeled for semantic equivalence; (2) RTE [19]: Sentence pairs for entailment prediction; (3) CoLA [84]: Sentences labeled for grammaticality; (4) SST-2 [73]: Sentences labeled for sentiment.

# E  Baseline details

This section provides a detailed baseline description. Our experiments encompass seven comparison methods:

- **Individual** means that each task uses an independent fine-tuned model, which has no interference between tasks, but cannot perform multiple tasks simultaneously.
- **Traditional MTL** collects the original training data of all tasks together to train a multi-task model. It can be used as a reference *upper bound* for model merging work.
- **Weight Averaging** is the simplest method of model merging, which directly averages the parameters of multiple models using $\theta_m = \sum_{t=1}^{n} \theta_t/n$, calculating the element-wise mean of all individual models. It can be used as a *lower bound* for model merging. [8, 86].
- **Fisher Merging** [46] calculates the Fisher information matrix [17] $\hat{F}_t = \mathbb{E}_{x \sim D_t} \mathbb{E}_{y \sim p_{\theta_t}(y|x)} \nabla_{\theta_t} (\log p_{\theta_t}(y|x_t))^2$ to measure the importance of each parameter when merging models for task $t$, where and model merging is performed according to the guidance of this importance.
- **RegMean** [30] imposes a constraint when merging models, that is, the $L_2$ distance between the merged model's and the individual models' activations. It computes a least-squares solution as $\theta_m = (\sum_{t=1}^{n} X_t^T X_t)^{-1} \sum_{t=1}^{n} (X_t^T X_t \theta_t)$, where $X_t$ is the input activation of the corresponding layer.
- **Task Arithmetic** [28] first defines the concept of "task vectors" and merges these vectors into a pre-trained model to execute multi-task learning. The model is produced by scaling and adding the task vectors to the initial model as $\theta_m = \theta_{\text{init}} + \lambda * \sum_{t=1}^{n} \tau_t$.
- **Ties-Merging** [89] further solves the task conflict problem in Task Arithmetic [28]. It eliminates redundant parameters and resolves symbol conflicts through three steps: Trim, Elect Sign, and Disjoint Merge.
- **AdaMerging** automatically learns a merging coefficient for each layer of each task vector in Task Arithmetic [28].
- **LoraHub** [25] employs Low-rank Adaptations to dynamically combine task-specific modules for cross-task generalization, and adapts to new tasks by configuring $\theta' = \sum_{k=1}^{K} w_k \cdot \theta_k$.
- **DARE** [94] sets the majority of delta parameters to zero and rescale the rest by $\theta' = \theta \cdot (1/(1-p))$ where $p$ is the proportion of delta parameters dropped, therefore efficiently reduces parameter redundancy.

# F  Implementation details

## F.1  Computational Resources and Runtimes

Our experiments were conducted on Nvidia A6000 GPUs with 48GB of RAM. Depending on the dataset size, fine-tuning the T5-Base and T5-Large models for single tasks took between 15 minutes and 2 hours, while fine-tuning the multitask checkpoint took around eight hours. The fine-tuned (IA)$^3$ models were provided by Yadav et al. [89].[6] We also used vision models ViT-B/32 and ViT-L/14 as provided by Ilharco et al. [28].[7]

Merge experiments were highly efficient, with evaluations for RoBerta-base, T5-Base, T5-Large, ViT-B/32, and ViT-L/14 models taking less than 2 minutes. However, two specific experiments required more time: (1) Evaluating (IA)$^3$ models took about one hour for 11 datasets due to the need to use multiple templates from prompt sources and compute median results across them. (2) Validation on LLMs (LLaMa2) was also slow, usually requiring about 40 minutes for evaluating 3 datasets.

## F.2  Training details

**Cross-Task Merging**  We trained the T5-base and T5-large models for up to 75,000 steps, using an effective training batch size of 1024 and a learning rate of 0.0001. To prevent overfitting, we implemented an early stopping mechanism with a patience of 5. Training was conducted in bfloat16 to

---

[6]https://github.com/prateeky2806/ties-merging
[7]https://github.com/mlfoundations/task_vectors#checkpoints

conserve GPU memory, with a maximum sequence length of 128 tokens. For the PEFT configuration of the (IA)$^3$ approach on the T0-3B model, we adjusted the parameters accordingly. The training batch size was set at 16, and the evaluation batch size was 32, while keeping the learning rate at 0.0001. Given the increased complexity, we extended the early stopping patience to 10. No learning rate scheduler or weight decay was used in any of our training processes. For large language models, we directly utilized the fine-tuned checkpoints provided by Huggingface[8].

**Cross-Domain Merging**   We performed fine-tuning of the RoBERTa-base model starting with an initial learning rate of 1e-5, and for the T5-base model, we used an initial learning rate of 1e-4. We applied the AdamW optimizer consistently across all experiments. The learning rate was set to gradually increase during the first 6% of training steps and then linearly decreased to zero. The models were trained with a batch size of 16 over 30 epochs for the task of emotion classification. We assessed model performance at the end of each epoch and, upon completing the training, resumed from the best-performing checkpoint.

**Cross-Training Configurations Merging**   When merging multiple checkpoints of the same task, each model is fine-tuned 10 times on each dataset using a random hyperparameter search. The learning rate is randomly selected in log space from $[10^{-6}, 10^{-3}]$, the batch size from $\{8, 16, 32, 64\}$, and the number of epochs from $\{2, 3, 5\}$. Evaluation occurs once at the end of training without early stopping. We use a maximum sequence length of 128 tokens and train the models using the Adam optimizer [33], with $\beta_1 = 0.9$, $\beta_2 = 0.999$ and $\epsilon = 10^{-8}$. Training includes gradient clipping at 1.0, no weight decay, and a learning rate that linearly decays to zero by the end of the process.

### F.3  Hyper-parameter settings

Given the sensitivity of task vector-based model merging methods to hyperparameters, we present the optimal values of $\lambda$ and $r$ as determined in our experiments, as shown in Tab. 17. For Task Arithmetic, we conduct a search over $\lambda$ ranging from 0.2 to 1.5 with a step size of 0.1. For TIES-Merging and PCB-MERGING, we search over mask ratios $r$ in {0.05, 0.1, 0.2}, and $\lambda$ ranging from 0.8 to 2.5 with a step size of 0.1.

Table 17: Optimal $\lambda$ and mask ratio $r$ for cross-task merging

| Task ($\rightarrow$) | 7 **NLP** Tasks | | 11 **PEFT** Tasks | 3 **LLM** Tasks | 8 **Vision** Tasks | |
| --- | --- | --- | --- | --- | --- | --- |
| Method ($\downarrow$) | T5-Base | T5-Large | (IA)$^3$ | LLaMa2 | ViT-B/32 | ViT-L/14 |
| Task Arithmetic[ICLR23] [28] [$\lambda$] | 0.4 | 0.5 | 0.5 | 0.3 | 0.3 | 0.3 |
| Ties-Merging[NeurIPS23] [89] [$\lambda, r$] | [1.7, 0.1] | [2.4, 0.05] | [1.7, 0.1] | [1.0, 0.1] | [1.0, 0.1] | [1.1, 0.05] |
| PCB-MERGING (ours) [$\lambda, r$] | [1.9, 0.05] | [2.2, 0.05] | [1.8, 0.1] | [0.9, 0.1] | [1.2, 0.05] | [1.2, 0.05] |

---

[8]https://huggingface.co/

