# OpenReview forum: "Parameter Competition Balancing for Model Merging"
_NeurIPS.cc/2024/Conference — NeurIPS 2024 poster_

### Official Review · Reviewer_Zv3Q · 2024-07-06

**Soundness:** 3
**Presentation:** 3
**Contribution:** 2
**Rating:** 6
**Confidence:** 4

**Summary:**

They propose a PCB for merging which computes parameter importance using 3 steps for intra-balancing and inter-balancing, which outperforms previous methods.

**Strengths:**

- Paper well structured and easy to follow
- Method outperforms baselines

**Weaknesses:**

- Lack of novelty - similar to TIES with some modified way to compute importance. Experimental setup also similar.
- No theoretical motivation for method

**Questions:**

- How important is the CMA-ES for hyperparameter tuning?
- What happened to Fisher Merging and RegMean for the 3 LLM Tasks in Table 2?
- Is the softmax done across all parameters per task vector? What is the Norm in eq 1?
- Is the number of searches done with a validation set the same across different methods in the results?
- In eq 2, should j exclude i (i.e. inter-balance is not computed between a model and itself)

---

> ### Author Rebuttal · Authors · 2024-08-07
>
> **Reply to Reviewer Zv3Q** \
> Thank you for your valuable comments. We will explain your concerns point by point.
>
> **Weaknesses 1：About the novelty.** \
>    **Reply**: Please refer to the second point in our general response document.
>
> **Weaknesses 2：About the theoretical motivation.** \
>    **Reply**: Please refer to the first point in our general response document.
>
> **Question 1: About the function of CMA-ES.** \
>    **Reply**: In fact, the function of CMA-ES is not to adjust the overall scaling hyperparameter $\lambda$. It is used to further search for the coefficients $\lambda_i$ for each task, as shown in Equation 5 in Section 3.3 of our paper. Typically, we use a uniform $\lambda$ as the initialization value for each model's $\lambda_i$, and then employ evolutionary strategies (ES) to search for a more accurate $\lambda_i$. CMA-ES is used to accelerate this search process. Recently, evolutionary strategies have also been widely used to enhance model merging in works like *Lorahub* [8], *EvoLLM* [9], and *Model_Evolver* [10].
>
> **Question 2: Fisher Merging and RegMean for the 3 LLM Tasks in Table 2.** \
>    **Reply**: The methods of Fisher Merging and RegMean are actually not suitable for LLMs. Firstly, these methods require more GPU resources, as they necessitate the additional computation of the Fisher Information Matrix or Inner Product Matrix. Specifically, the RegMean method is not feasible even with GPUs having 80GB of memory for 7B LLMs. Moreover, both methods are gradient-based, making the results highly sensitive to the choice of small sample data and the number of training iterations, which significantly increases the complexity and instability of LLM merging. Considering these factors, we primarily compare methods based on task vectors for LLM merging, as they are easier and more lightweight to implement.
>
> **Question 3: About softmax and Norm.** \
>    **Reply**: It is true that the softmax is done across all parameters per task vector. Norm refers to the normalization of the task vector, which enhances numerical stability. The choice of softmax is largely due to the exponential function it incorporates, which amplifies larger values and diminishes smaller ones, thus increasing the contrast. This helps in converting the output vector into a probability distribution. In our paper, Appendix B.1 (Additional Ablation Studies), we replaced the softmax activation function with common alternatives like sigmoid, ReLU, and tanh. The results show minimal performance loss with different activation functions. This is because these activation functions can represent complex nonlinear relationships to balance the values of parameters.
>
> **Question 4: About the number of searches.** \
>    **Reply**: Please refer to the fourth point in our general response and **table 1 in PDF file in general response**.
>
> **Question 5: Should j exclude i.** \
>    **Reply**: We considered this issue at the initial design stage of our experiment. In our early experiments with merging eight models, we found that whether or not to exclude the model itself when computing inter-balance had a negligible impact on the results. Therefore, we opted for a simpler approach by not excluding it, which made the process straightforward and the formulas more concise.
>    In our current experiments merging three LLM tasks, we found that excluding the model itself resulted in a score of 35.14, which is a slight improvement compared to the original score of 35.12. This suggests that this issue has minimal impact on our method. The term "inter-balance" can be understood as the balance between the model and the entire population, or the balance among the other individuals within the population.
>
>  We thank the Reviewer again for the useful comments. We have revised the manuscript according to the Reviewer’s suggestion and response to each comment provided in the Weakness section above. We hope that our rebuttal aligns with the reviewer’s expectations, and we hope that the Reviewer can consider possibly giving a higher rating. Thanks.
>
> References: \
>    [8] Huang et al. LoraHub: Efficient cross-task generalization via dynamic LoRA composition. (arXiv23) \
>    [9] Akiba et al. Evolutionary optimization of model merging recipes. (arXiv24) \
>    [10] Du et al. Knowledge fusion by evolving weights of language models. (ACL24) \
>    [11] Yu et al. Language models are super mario: Absorbing abilities from homologous models as a free lunch. (ICML24)

---

> > ### Comment · Reviewer_Zv3Q · 2024-08-12
> >
> > Thanks for the response. I keep my score as is.

---

### Official Review · Reviewer_s8Ho · 2024-07-09

**Soundness:** 3
**Presentation:** 3
**Contribution:** 4
**Rating:** 7
**Confidence:** 4

**Summary:**

This paper introduces an innovative technique named PCB-MERGING (Parameter Competition Balancing), a lightweight and training-free technique that adjusts the coefficients of each parameter for effective model merging.

**Strengths:**

1. This paper re-examines existing model merging methods, highlighting the critical role of parameter competition awareness;
2. This paper introduce a novel approach called PCB-MERGING, which effectively adjusts parameter coefficients through balancing parameter competition;
3. The method stabilizes and enhances model merging performance across various application scenarios without additional training.

**Weaknesses:**

1. Figure 1 and 2 need to be re-explained. The meaning of the percentage is confusing. Does it refer to the pruning ratio or the proportion of adjusted key parameters? The meaning of scale also needs to be re-explained.
2. The time complexity in MOEA is a crucial topic for discussion. If additional MOEA-related algorithms are introduced, a time complexity analysis needs to be conducted.
3. An important mathematical symbols have not been defined, circle with dot.
4. The code seems unavailable? I'm curious about the process of selecting the task vector.

**Questions:**

Described in weaknesses. I might improve the rating if my questions are well addressed.

**Limitations:**

Described in weaknesses. I might improve the rating if my questions are well addressed.

---

> ### Author Rebuttal · Authors · 2024-08-07
>
> **Reply to Reviewer s8Ho** \
> Thank you for your valuable comments. We will explain your concerns point by point.
>
> **Weaknesses 1：More details in Figure 1 and 2** \
>    **Reply**: The term 'magnitude' refers to the magnitude of the task vector in Fig. 1. We will clarify this in the introduction of our final version by adding explanatory details. Thank you for your suggestions. Figure 1 illustrates an interesting phenomenon: scaling the top percentiles of a task vector outperforms fine-tuning, which is consistent with the ideas presented in the paper *DARE* [11]. \
>    The percentage represents the top magnitude percentiles of a task vector. It pertains to both the pruning ratio $r$ and the proportion of adjusted key parameters shown in Fig. 2, reflecting different representations in drop and rescale operations. However, in the later part where we introduce our new method, PCB-Merging, we first adjust all parameters and then perform pruning based on these adjustment results.
>
> **Weaknesses 2：Time complexity in MOEA** \
>    **Reply**: The total time required for the overall evolutionary strategy is $T_{\text{total}} = \text{Generations} \times (T_{\text{merging}} + T_{\text{validate}})$, where "Generations" represents the number of generations needed for evolution. The time required for model merging primarily depends on the number of model parameters and the size of the model population, while the time for model validation is mainly influenced by the volume of inference data and inference speed. We have compiled and reported the number of generations and the time required for each task in **Table 1 of the general response PDF**, and we analyze these factors in the fourth point of the general response. Recently, evolutionary strategies have also been widely used to enhance model merging in works like *Lorahub* [8], *EvoLLM* [9], and *Model_Evolver* [10].
>
> **Weaknesses 3：Definition of circle with dot** \
>    **Reply**: We interpret ⊙ as an element-wise product and use Norm as a shorthand for normalization. For more details, please refer to the third point in our general response document.
>
> **Weaknesses 4：Code for processing the task vector** \
>    **Reply**: Actually, we have provided the source code in the supplemental material **.zip file** during our initial submission. The details of our method are shown in the file `pcb-merging.py`. Additionally, you can check the application in different scenarios with evolutionary strategies in `pcb_ES.py` (found in the `vision_source_code` directory) or `merging.py` (in the `nlp_source_code` directory). You can obtain the model population by executing `run_finetuning.sh` and try different merging methods using `run_merging.sh`.
>
>  We thank the Reviewer again for the useful comments. We have revised the manuscript according to the Reviewer’s suggestion and response to each comment provided in the Weakness section above. We hope that our rebuttal aligns with the reviewer’s expectations, and we hope that the Reviewer can consider possibly giving a higher rating. Thanks.
>
> References: \
>    [8] Huang et al. LoraHub: Efficient cross-task generalization via dynamic LoRA composition. (arXiv23) \
>    [9] Akiba et al. Evolutionary optimization of model merging recipes. (arXiv24) \
>    [10] Du et al. Knowledge fusion by evolving weights of language models. (ACL24) \
>    [11] Yu et al. Language models are super mario: Absorbing abilities from homologous models as a free lunch. (ICML24)

---

### Official Review · Reviewer_28gt · 2024-07-12

**Soundness:** 3
**Presentation:** 2
**Contribution:** 3
**Rating:** 6
**Confidence:** 4

**Summary:**

The authors propose an improved method to merge task vectors, called Parameter Competition Balancing (PCB-merging). The proposed method is simple, efficient, and attains superior performance in evaluations.

**Strengths:**

1. In Figure 1, the authors show a very interesting phenomena, where scaling the top percentiles of a task vector outperforms fine tuning.
2. The proposed method is very simple. It can be implemented in a few lines of code, requires very little memory or computation, and does not require additional data. These are crucial advantages that make PCB-merging highly practical when also considering its improved performance.
3. The method performs well, surpassing TIES-merging, Task Arithmetic, RegMean, and standard averaging in a variety of settings.

**Weaknesses:**

1. The paper has numerous typos.

**Questions:**

1. In the introduction for Figure 1, it's not quite clear what "Magnitude" refers to. Is it the magnitude in the task vector or the magnitude after the task vector has been applied? This is answered later but it would be helpful to add a word or two to clear this up in the introduction.
2. I am a bit confused about the actual operations used for the parameter balancing. I am interpreting ⊙ as an element-wise product. Thus τᵢ ⊙ τᵢ is a vector and Norm(τᵢ ⊙ τᵢ) would be a scalar, but Softmax requires a vector. It might be helpful to clarify the notation in Section 3.2 a bit more.

**Limitations:**

1. The authors point out that the mechanism leading to the improved performance is not well understood.
2. The merging method likely cannot be used to merge models that are fine tuned from different pretrained models.

I believe both of these limitations are acceptable.

---

> ### Author Rebuttal · Authors · 2024-08-07
>
> **Reply to Reviewer 28gt** \
> Thank you for your valuable comments. We will explain your concerns point by point.
>
> **Weaknesses 1：More details in Figure 1.** \
>    **Reply**: The term 'magnitude' refers to the magnitude of the task vector in Fig. 1. We will clarify this in the introduction of our final version by adding explanatory details. Thank you for your suggestions. Figure 1 illustrates an interesting phenomenon: scaling the top percentiles of a task vector outperforms fine-tuning, which is consistent with the ideas presented in the paper *DARE* [11].
>
> **Weaknesses 2：Clarification regarding the notation for Softmax and Norm in Section 3.2.** \
>    **Reply**: Please refer to the third point in our general response document.
>
> **Limitation 1：Theoretical understanding.** \
>    **Reply**: Please refer to the first point in our general response document.
>
> **Limitation 2：Reliance on shared initializations for applications.** \
>    **Reply**: Please refer to the second point in our general response document. Shared initializations are a fundamental issue in model merging, and more complex scenarios can be addressed by converting them into shared initializations through methods like *FuseLLM* [7]. Common applications include multi-task learning [1], multi-domain adaptation [2], merging various training strategies [3], model compression [4], and mitigating catastrophic forgetting [5], among others.
>
> We thank the Reviewer again for the useful comments. We have revised the manuscript according to the Reviewer’s suggestion and response to each comment provided in the Weakness section above. We hope that our rebuttal aligns with the reviewer’s expectations, and we hope that the Reviewer can consider possibly giving a higher rating. Thanks.
>
> References: \
>    [1] Ilharco et al. Editing models with task arithmetic. (ICLR 2023)  \
>    [2] Jin et al. Dataless knowledge fusion by merging weights of language models. (ICLR 2023)  \
>    [3] Yadav et al. Ties-merging: Resolving interference when merging models. (NeurIPS 2023)  \
>    [4] Wang et al. Localizing task information for improved model merging and compression. (ICML24) \
>    [5] Zhu et al. Model tailor: Mitigating catastrophic forgetting in multi-modal large language models. (ICML24) \
>    [7] Wan et al. Knowledge fusion of large language models. (ICML24) \
>    [11] Yu et al. Language models are super mario: Absorbing abilities from homologous models as a free lunch. (ICML24)

---

> > ### Comment · Reviewer_28gt · 2024-08-12
> >
> > I would like to thank the authors for their rebuttal. My concerns have been addressed, but since I did not note any major issues to begin with, I will keep my score.

---

### Official Review · Reviewer_6PSY · 2024-07-18

**Soundness:** 3
**Presentation:** 4
**Contribution:** 3
**Rating:** 5
**Confidence:** 5

**Summary:**

This paper focuses on the model merging problem. The authors propose PSC-MERGING to adjust the coefficients of each parameter for effective model merging. Specifically, PSC-MERGING uses intra-balancing to weight the importance of parameters within tasks and inter-balancing to assess parameter similarities across tasks. Parameters with low importance scores are dropped, and the remaining ones are rescaled to form the final merged model. The authors conduct extensive experiments to validate the proposed method.

**Strengths:**

+ The paper is well organized. The motivation is well provided and the authors make a comprehensive survey on the related works.

+ The authors conduct extensive experiments, including cross-task merging, cross-domain merging, cross-training configuration merging and out-of-domain generalization, to validate the proposed method. The proposed method shows clear superiority in all these settings.

**Weaknesses:**

+ The proposed method introduces an hyperparameter r in Eqn. (3). From Figure 6, it can be seen that the performance is highly sensitive to this hyperparameter, which may make the proposed method not easy to use in practice. Furthermore, the authors claim “For TIES-Merging and PCB-MERGING, which require a masking ratio, we set mask ratio r = 0.2 as the default value for all experiments, except in LLM experiments where r = 0.1.” However, From Figure 6 it is obviously that the optimal r should be 0 instead of 0.2. The setting of the hyperparameter r should be more justified.

+ Another concern is that the authors use quite different search space when setting the hyperparameter lambda, as described in Line #217~222. Why does the search range of this parameter vary so much under different methods? Will it incur any unfair comparison? The rationale for such settings should be provided.

+ The proposed method relies on shared initializations, which significantly limits its applicability as acknowledged by the authors.

**Questions:**

Please see the Weaknesses

---

> ### Author Rebuttal · Authors · 2024-08-07
>
> **Reply to Reviewer 6PSY** \
> Thank you for your valuable comments. We will explain your concerns point by point.
>
> **Weaknesses 1：The setting of the hyperparameter $r$.** \
>    **Reply**: The reviewer has some confusion and misunderstandings regarding our experiment settings.
>    Firstly, our hyperparameters are discussed in two scenarios: When no additional validation is performed, the masking ratio for TIES-Merging and PCB-Merging is set to $ r = 0.2 $. When validation is allowed, we search over ratios in $\{0.05, 0.1, 0.2\}$.
>    Additionally, in Figure 6, the optimal $ r $ is not 0; it should be 0.05 instead. This conclusion is consistent with our findings in Appendix F.3, Table 17, concerning hyperparameter settings. \
>    In fact, when $ r $ is close to 0, the performance of TIES-Merging and PCB-Merging drops sharply, as shown in **Figure 1 in the general response PDF** file. The optimal range for $ r $ is between 0.03 and 0.3. To improve feasibility, we only search within the values $\{0.05, 0.1, 0.2\}$.
>
> **Weaknesses 2：The rationale for the various settings of different methods.** \
>    **Reply**: In this paper, the selection of hyperparameters for different methods adheres to the original papers and is indeed suitable for obtaining optimal solutions for these methods. Therefore, this comparison is fair and reasonable. Specifically: \
>    For *TIES-Merging*, we followed the hyperparameter settings outlined in Appendix C.4 and C.5 of the original paper. \
>    For *Task Arithmetic*, we adhered to the hyperparameter settings in Appendix D (Learning via addition) of the original paper. \
>    For *RegMean*, we followed the discussion on the impact of scaling non-diagonal values in Inner Product Matrices in section 5.3 of the original paper. \
>    Lastly, since the meaning of the hyperparameter $\lambda$ varies across different methods, we summarized the range and optimal values of hyperparameters for each method in our paper. This is detailed in the Experimental Setup section and Appendix F.3, **Table 17**.
>
> **Weaknesses 3：Reliance on shared initializations for applications.**: \
>    **Reply**: Please refer to the second point in our general response document. Shared initializations are a fundamental issue in model merging, and more complex scenarios can be addressed by converting them into shared initializations through methods like *FuseLLM* [7]. Common applications include multi-task learning [1], multi-domain adaptation [2], merging various training strategies [3], model compression [4], and mitigating catastrophic forgetting [5], among others.
>
> We thank the Reviewer again for the useful comments. We have revised the manuscript according to the Reviewer’s suggestion and response to each comment provided in the Weakness section above. We hope that our rebuttal aligns with the reviewer’s expectations, and we hope that the Reviewer can consider possibly giving a higher rating. Thanks.
>
> References: \
>    [1] Ilharco et al. Editing models with task arithmetic. (ICLR 2023)  \
>    [2] Jin et al. Dataless knowledge fusion by merging weights of language models. (ICLR 2023)  \
>    [3] Yadav et al. Ties-merging: Resolving interference when merging models. (NeurIPS 2023)  \
>    [4] Wang et al. Localizing task information for improved model merging and compression. (ICML24) \
>    [5] Zhu et al. Model tailor: Mitigating catastrophic forgetting in multi-modal large language models. (ICML24) \
>    [7] Wan et al. Knowledge fusion of large language models. (ICML24)

---

> > ### Comment · Reviewer_6PSY · 2024-08-13
> >
> > I appreciate author's feedback. Some of my concern are resolved. I keep my original positive score.

---

> > > ### Author Response · Authors · 2024-08-13
> > > **Replying to Reviewer 6PSY**
> > >
> > > Thank you for your positive score and insightful feedback. If you have any concerns about our work, we would greatly appreciate receiving any further comments or suggestions.

---

### Author Rebuttal · Authors · 2024-08-07

**General Response**
We appreciate your consideration in taking the time to review our comments. We have received feedback from four reviewers, all of whom have provided thoughtful insights. Almost all reviewers found our paper to be well-organized, motivated, practical, and easy to follow. Additionally, our paper has demonstrated extensive experiments and applications, leading to improved performance. However, there are still some questions and concerns from the reviewers. We have summarized and addressed these in four key points.

1. **Motivation and theoretical understanding**:  \
   Our research aims to improve the performance of model merging by addressing parameter competition through a balancing mechanism that adjusts parameter-level coefficients. Our method, PCB-Merging, retains the advantages of being lightweight, easy to implement, and high-performing. We systematically compare and analyze existing model merging methods regarding their intra-balancing and inter-balancing capabilities, emphasizing the importance of parameter competition awareness. We establish a balancing matrix that is self-aware and cross-aware for parameter scaling. Furthermore, we introduce PCB-Merging, a novel approach that effectively adjusts parameter coefficients by balancing parameter competition.

2. **Application scenarios of our method**:  \
   Firstly, as with most current research, model merging methods are limited by shared initializations. Despite this, they have a wide range of applications, such as multi-task learning [1], multi-domain adaptation [2], merging various training strategies [3], and model compression [4]. Secondly, our method can also enhance transferability and mitigate catastrophic forgetting. For instance, merging the transferred model with the original model in pairs can improve practicality, as demonstrated in the papers *Model Tailor* [5] and *Robust Fine-tuning* [6]. Finally, in knowledge fusion scenarios, we can use methods like knowledge distillation and model alignment to convert models with different initializations to the same initializations. This approach can even be applied across different frameworks to achieve model merging, as shown in the paper *FuseLLM* [7].

3. **Clarification regarding the notation in Section 3.2**: \
   We apologize for any confusion regarding our use of the term "Norm" in the manuscript. We meant the normalization of the vector τi ⊙ τi, not its norm (which would be a scalar). Normalizing this vector retains its vector form.
   Softmax is applied across all parameters per task vector. Normalizing the task vector improves numerical stability. We chose softmax for its exponential function, which increases contrast by amplifying larger values and diminishing smaller ones, converting the output into a probability distribution.
   In Appendix B.1 (Additional Ablation Studies), we replaced softmax with alternatives like sigmoid, ReLU, and tanh. The results showed minimal performance loss, as these functions also balance parameter values effectively.

4. **Time Complexity for Evolution Strategy**: \
   Recently, evolutionary strategies have also been widely used to enhance model merging in works like *Lorahub* [8], *EvoLLM* [9], and *Model_Evolver* [10]. We propose evolution strategy to search for the coefficients $\lambda_i$ for each task, as shown in Equation 5 in Section 3.3 of our paper. A specific CMA-ES algorithm is used to accelerate the search process.
   The total time required for the overall evolutionary strategy is $T_{\text{total}} = \text{Generations} \times (T_{\text{merging}} + T_{\text{validate}})$, where "Generations" represents the number of generations needed for evolution. The time for model merging mainly depends on the number of model parameters and the size of the model population, while the time for model validation primarily depends on the volume of inference data and the inference speed.  We have organized and reported the number of generations and the time required for each task in Table 1 of the general response PDF file, and analyzed this in the fifth point of the general response.

Once again, we sincerely thank you for your involvement and thoughtful feedback! We have provided detailed responses to each question from each reviewer. Additionally, we have included the source code and the experimental procedures for the evolution strategy in NLP, vision, and other applications in the supplemental material zip file for your reference.

References: \
   [1] Ilharco et al. Editing models with task arithmetic. (ICLR 2023)  \
   [2] Jin et al. Dataless knowledge fusion by merging weights of language models. (ICLR 2023)  \
   [3] Yadav et al. Ties-merging: Resolving interference when merging models. (NeurIPS 2023)  \
   [4] Wang et al. Localizing task information for improved model merging and compression. (ICML24) \
   [5] Zhu et al. Model tailor: Mitigating catastrophic forgetting in multi-modal large language models. (ICML24) \
   [6] Wortsman et al. Robust fine-tuning of zero-shot models. (CVPR22) \
   [7] Wan et al. Knowledge fusion of large language models. (ICML24) \
   [8] Huang et al. LoraHub: Efficient cross-task generalization via dynamic LoRA composition. (arXiv23) \
   [9] Akiba et al. Evolutionary optimization of model merging recipes. (arXiv24) \
   [10] Du et al. Knowledge fusion by evolving weights of language models. (ACL24)

---

### Decision · Program_Chairs · 2024-09-25

**Decision:**

Accept (poster)

**Comment:**

Initially, the paper received four positive scores, no major concerns, yet writing is a relatively common concern among the reviewers. After the rebuttal, the authors solved some of the concerns from the reviewers, and the reviewers maintained the positive scores. The AC read through the manuscript, all reviews, the discussion, and the rebuttal. The authors are highly encouraged to improve the paper quality according to the reviewers' feedback in the camera-ready version. The AC decided to accept this submission.